# γ-TuRC asymmetry induces local protofilament mismatch at the RanGTP-stimulated microtubule minus end

Bram JA Vermeulen [ID][1,4], Anna Böhler[1,4], Qi Gao[1,4], Annett Neuner[1], Erik Župa[1], Zhenzhen Chu[2,3], Martin Würtz [ID][1], Ursula Jäkle[1], Oliver J Gruss [ID][2], Stefan Pfeffer [ID][1✉] & Elmar Schiebel [ID][1✉]

## Abstract

The γ-tubulin ring complex (γ-TuRC) is a structural template for de novo microtubule assembly from α/β-tubulin units. The isolated vertebrate γ-TuRC assumes an asymmetric, open structure deviating from microtubule geometry, suggesting that γ-TuRC closure may underlie regulation of microtubule nucleation. Here, we isolate native γ-TuRC-capped microtubules from *Xenopus laevis* egg extract nucleated through the RanGTP-induced pathway for spindle assembly and determine their cryo-EM structure. Intriguingly, the microtubule minus end-bound γ-TuRC is only partially closed and consequently, the emanating microtubule is locally misaligned with the γ-TuRC and asymmetric. In the partially closed conformation of the γ-TuRC, the actin-containing lumenal bridge is locally destabilised, suggesting lumenal bridge modulation in microtubule nucleation. The microtubule-binding protein CAMSAP2 specifically binds the minus end of γ-TuRC-capped microtubules, indicating that the asymmetric minus end structure may underlie recruitment of microtubule-modulating factors for γ-TuRC release. Collectively, we reveal a surprisingly asymmetric microtubule minus end protofilament organisation diverging from the regular microtubule structure, with direct implications for the kinetics and regulation of nucleation and subsequent modulation of microtubules during spindle assembly.

**Keywords** Gamma-tubulin; Ring Complex; Microtubule Nucleation; RanGTP; Cryo-EM
**Subject Categories** Cell Adhesion, Polarity & Cytoskeleton; Structural Biology

## Introduction

Microtubules (MTs) are cytoskeletal filaments central to numerous cellular processes in eukaryotes, including spindle formation, cellular trafficking, cell polarisation and force generation. MTs are composed of laterally associated protofilaments of α/β-tubulin dimers with a fast-growing plus end and less dynamic minus end. MTs can spontaneously assemble in vitro, resulting in varying protofilament numbers. In vivo, however, MT nucleation is templated by γ-tubulin complexes (γ-TuCs), allowing for spatio-temporal regulation of MT nucleation and subsequent MT minus-end anchoring (Wiese and Zheng, 2000). In addition, this potentially enables γ-TuCs to determine the protofilament number of the nucleated MT, which is 13 in the vast majority of situations (Chaaban and Brouhard, 2017; Tilney et al, 1973).

In eukaryotes, the γ-tubulin ring complex (γ-TuRC) functions as the major MT nucleator (Bohler et al, 2021), which is essential for cell viability (Funk et al, 2022). In interphase, MT nucleation by the γ-TuRC is activated by CDK5RAP2 (CDK5 regulatory subunit-associated protein 2) and primarily occurs at the centrosome (Choi et al, 2010). However, the bulk of MT nucleation occurs in a CDK5RAP2- and centrosome-independent manner during the assembly of mitotic and meiotic spindles (Petry, 2016; Tariq et al, 2020). A major pathway for MT-nucleation in M-phase is chromatin-dependent, in which the presence of RanGTP in the vicinity of chromosomes activates spindle assembly factors (SAFs), including TPX2 (Gruss et al, 2001), Augmin (Kraus et al, 2023; Ustinova et al, 2023) and HURP (Hayward et al, 2014; Silljé et al, 2006). These SAFs mediate branching MT nucleation (Gruss et al, 2001; Petry et al, 2013; preprint: Valdez et al, 2023), in which new MTs are nucleated from the side of existing MTs by the γ-TuRC. This pathway has been extensively studied in the vertebrate *Xenopus laevis* meiotic egg extract system, a cytoplasmic extract in which spindle-like structures readily form upon the addition of RanGTP (Carazo-Salas et al, 1999; Ohba et al, 1999; Wilde and Zheng, 1999).

For many years since the discovery of γ-tubulin (Oakley and Oakley, 1989), the structure of the γ-TuRC remained elusive. Medium-resolution cryo-EM reconstructions of the compositionally much simpler budding yeast γ-tubulin small complex (γ-TuSC) formed the basis for understanding the structure of γ-TuCs (Kollman et al, 2015; Kollman et al, 2010). These analyses revealed a spoked assembly in which the related γ-tubulin complex proteins GCP2 and GCP3 alternately present γ-tubulin in an arrangement

[1]Zentrum für Molekulare Biologie, Universität Heidelberg, DKFZ-ZMBH Allianz, Heidelberg, Germany. [2]Institut für Genetik, Universität Bonn, Bonn, Germany. [3]Present address: Key Laboratory of Carcinogenesis and Translational Research (Ministry of Education, Beijing), Lymphoma Department, Peking University Cancer Hospital & Institute, Beijing, China. [4]These authors contributed equally: Bram JA Vermeulen, Anna Böhler, Qi Gao. ✉E-mail: s.pfeffer@zmbh.uni-heidelberg.de; e.schiebel@zmbh.uni-heidelberg.de

similar to the helical parameters of 13-protofilament MTs (Kollman et al, 2015; Kollman et al, 2010), providing strong evidence for the long-standing hypothesis that γ-TuCs serve as a structural template for MT nucleation by mimicking one layer of α/β-tubulin dimers. Recently, the structure of the vertebrate γ-TuRC was resolved at high resolution using cryo-EM (Consolati et al, 2020; Liu et al, 2020; Wieczorek et al, 2020b; Zimmermann et al, 2020), providing insights into the role of γ-TuRC-specific components GCP4-6 as well as mitotic spindle organizing proteins (MZT) 1 and 2 (Wieczorek et al, 2020a). The vertebrate γ-TuRC assembles into a uniform 14-spoked arrangement, which is lined on the inside by a lumenal bridge composed of actin and two structural modules formed by MZT1 and the N-termini of GCP3 and GCP6. MZT2, on the other hand, forms a structural module with the N-terminus of GCP2, which has thus far only been observed in cryo-EM reconstructions when stably docked to the γ-TuRC in complex with the CM1 motif of CDK5RAP2 (Wieczorek et al, 2020a; Wieczorek et al, 2020b; preprint: Xu et al, 2023). Crosslinks from the MZT2/GCP2$^N$ module map to the outside surface of GCP2/3 (Zimmermann et al, 2020), suggesting that this module primarily samples positions on the outside of the γ-TuRC even in the absence of CDK5RAP2.

In contrast to the oligomerised budding yeast γ-TuSC, which can form a perfect template for MT nucleation (Brilot et al, 2021; Kollman et al, 2015), all vertebrate γ-TuRC structures resolved to date display an 'open' conformation that markedly deviates from MT geometry (Consolati et al, 2020; Liu et al, 2020; Wieczorek et al, 2020b; Wurtz et al, 2022; Zimmermann et al, 2020), especially in the terminal four spokes (10–14), compromising its function as a structural template for MT formation. Consistently, the asymmetric γ-TuRC is an inefficient MT nucleator in vitro (Consolati et al, 2020; Thawani et al, 2020). Based on the imperfect helical geometry of the γ-TuRC, it has been proposed that the γ-TuRC may undergo conformational changes in the context of MT nucleation (Consolati et al, 2020; Liu et al, 2020; Thawani et al, 2020; Wieczorek et al, 2020b; Zupa et al, 2021). Lateral interactions of protofilaments in the assembling MT may passively close the γ-TuRC to MT-compatible geometry during the nucleation reaction (Thawani et al, 2020). Alternatively, factors that enhance MT nucleation by the γ-TuRC, such as CDK5RAP2 (Choi et al, 2010; Rale et al, 2022; preprint: Romer et al, 2023; preprint: Xu et al, 2023), could induce conformational closure of the γ-TuRC before or during MT nucleation and thereby enhance MT nucleation activity. Finally, the γ-TuRC may remain in an asymmetric conformation, which would have a profound impact on the structure of the MT-γ-TuRC interface and may create a specific recognition site for proteins regulating γ-TuRC turnover, as has been observed before (Rai et al, 2024). However, the absence of structural data on the γ-TuRC attached to the MT minus end during or after a MT nucleation event precludes mechanistic understanding of γ-TuRC activation and anchoring of MTs.

Here, we have isolated γ-TuRC-capped MTs from MT asters nucleated through the native RanGTP pathway in *X. laevis* egg extract and characterised the structure of the γ-TuRC at the MT minus end using cryo-EM. We show that the γ-TuRC unexpectedly adopts a partially closed conformation at the minus end, transitioning towards—but not fully reaching—MT symmetry. Strikingly, the associated MT protofilaments are conformationally misaligned with the γ-TuRC, suggesting that MT minus end

anchoring to the γ-TuRC is comparably loose. Furthermore, the emanating MT deviates substantially from MT lattice geometry, including a break in contacts between specific protofilaments. Notably, CAMSAP2 (calmodulin-regulated spectrin-associated protein 2) recognises the γ-TuRC-capped MT minus end, indicating that structural properties inherent to its asymmetry enable site-specific recruitment of MT-modulating factors. Lastly, the γ-TuRC defines the position of the seam in the nucleated MT, providing a possible mechanism for orienting the seam in MT-containing organelles. Cumulatively, this work reveals an inherent asymmetry in MT nucleation through the RanGTP pathway that has functional consequences for the regulation of MT nucleation kinetics and modulation of subsequent MT anchoring, advancing our mechanistic understanding of this essential cellular process.

## Results

### Purification of natively nucleated γ-TuRC-capped MT minus ends from *X. laevis* egg extract

To study the structure of the γ-TuRC after MT nucleation, we established a protocol to extract native γ-TuRC-capped MTs from meiotic *Xenopus laevis* egg extract (Appendix Fig. S1A). Structural analysis of γ-tubulin complexes from native sources was long hampered by its low cellular abundance (Beck et al, 2011) and purifying the γ-TuRC with attached MTs nucleated through a native pathway adds several additional layers of complexity to the purification scheme. Still, we chose this approach because it allows MT formation under native conditions, including endogenous wild-type α/β-tubulin as well as MT- and γ-TuRC-associated factors that may stimulate and regulate MT nucleation, all with correct post-translational modifications.

Briefly, MT assembly was induced in *X. laevis* egg extracts by the addition of GTPase-deficient Ran mutant Q69L (hereinafter referred to as RanGTP, Appendix Fig. S1B), which activates the chromosomal pathway for MT nucleation and spindle assembly (Carazo-Salas et al, 1999; Ohba et al, 1999) in a γ-TuRC-dependent manner (Wilde and Zheng, 1999). Only after MT asters were formed, nucleated MTs were stabilised using Paclitaxel (Taxol) or docetaxel (also known as Taxotere, henceforth referred to as DTX), pelleted and shortened by mild mechanical shearing to a length suitable for cryo-EM analysis. Finally, γ-TuRC-capped MTs were affinity purified using anti-γ-tubulin antibody-coupled beads. The presence of γ-TuRC-capped MTs was verified using negative stain EM (Appendix Fig. S2A). Consistently, mass spectrometry analysis identified α/β-tubulin as well as γ-TuRC components and interacting proteins in the sample (Appendix Table S1, full list available as source data), including components involved in RanGTP/Augmin-dependent MT nucleation, such as TPX2, XMAP215 and the Augmin subunits HAUS1-8. Consistent with the absence of centrosomes in *X. laevis* egg extract (Gruss, 2018), we did not find the CM1-containing protein CDK5RAP2, which activates centrosomal MT nucleation. This confirms that purified MT minus ends were nucleated in a native RanGTP-dependent spindle assembly pathway. Importantly, in the absence of RanGTP induction, the number of γ-TuRC-capped MTs observed using negative stain EM was reduced 32-fold for a highly active extract as

used in this study for cryo-EM analysis of γ-TuRC-capped MTs (Appendix Fig. S2A,B). This indicates that the vast majority of γ-TuRC-capped MTs analysed by cryo-EM were indeed nucleated through the RanGTP pathway and did not result from γ-TuRC binding to the end of MTs spontaneously nucleated by the MT stabiliser Paclitaxel.

## A tailored image processing scheme enables structural analysis of natively nucleated γ-TuRC-capped MTs

For initial structural analysis, we used natively nucleated γ-TuRC-capped MT minus ends stabilised using Paclitaxel. We acquired multiple independent cryo-EM datasets and manually identified candidate particles (Fig. 1A), which were subjected to an initial step of 2D classification for enrichment of structurally defined capped MT minus ends (Fig. 1B). Through analysis of the characteristic Moiré patterns of the MTs as well as computational particle sorting (Fig. 1C,D), we determined that >90% of the capped MTs retained after 2D classification were comprised of 13 protofilaments. Moreover, cryo-EM 3D reconstruction of the lattice of capped MTs away from the very minus end readily resolved ordered tubulin secondary structure, indicating that the structural integrity of MTs was maintained throughout the purification process (Appendix Fig. S3A–C).

Particle alignment of MT-capping γ-TuRCs was initially hampered by the overall limited number of particles, originating from the challenges of purification from a native source. The large mass and apparent symmetry of the associated MTs and the high structural similarity of the individual γ-TuRC spokes further complicated particle alignment (see Methods). We thus devised a tailored multistep approach for data processing (Fig. EV1A,B). In the first step, all particle images were aligned in 2D, defining their in-plane rotation as well as the position of the MT minus end. In the second step, particles were refined in 3D for initial centring of the particles, focusing only on

the γ-TuRC, masking out the MT minus end. During all 3D refinement steps, orientations were sampled globally around the MT axis to take into account the pseudohelical symmetry, whereas the other two rotation axes were sampled only locally around orientations found in the robust 2D alignment step. Due to the relatively low number of particles, the attainable level of detail in the initial refinement run was limited and density features indicated that the spoke register was not determined correctly for every particle. Therefore, we performed a subsequent round of refinement after supplementing the γ-TuRC-capped MT minus end particle images with particle images of isolated γ-TuRC from *Xenopus laevis* egg extract (Liu et al, 2020). This aided the alignment of MT-capping γ-TuRCs and thereby allowed the refinement to reach subnanometer resolution, sufficient for accurate particle alignment on individual GCP-γ-tubulin spokes (see Discussion). The strategy of particle supplementation had no detectable effect on the structure of the MT-capping γ-TuRC, as validated in a series of in silico experiments (Figs. EV1C and EV2A–C, Table EV1 and Appendix Table S2, see Discussion). Most importantly, all asymmetric structural features of the MT-capping γ-TuRC were retained in a refinement run without any supplemented particles, although at lower resolution (Fig. EV2A–C). Retaining only the capped MT minus end particles for the reconstruction, we obtained an initial cryo-EM density of the MT-capping γ-TuRC at 23 Å resolution (Fig. EV3A).

In this reconstruction, we observed a uniquely asymmetric MT protofilament organisation with locally divergent protofilament arrangements (see below). Hence, to validate that the structure of the γ-TuRC-capped MT minus end was not affected by Paclitaxel-induced changes in protofilament interactions, we repeated the structural analysis with γ-TuRC-capped MT minus ends stabilised after nucleation using the drug DTX, which is neutral to lateral interactions in the MT (Andreu et al, 1994; Diaz et al, 1998; Prota et al, 2023). Following the same processing scheme as above, we obtained a cryo-EM density of the MT-capping γ-TuRC at 23 Å resolution,

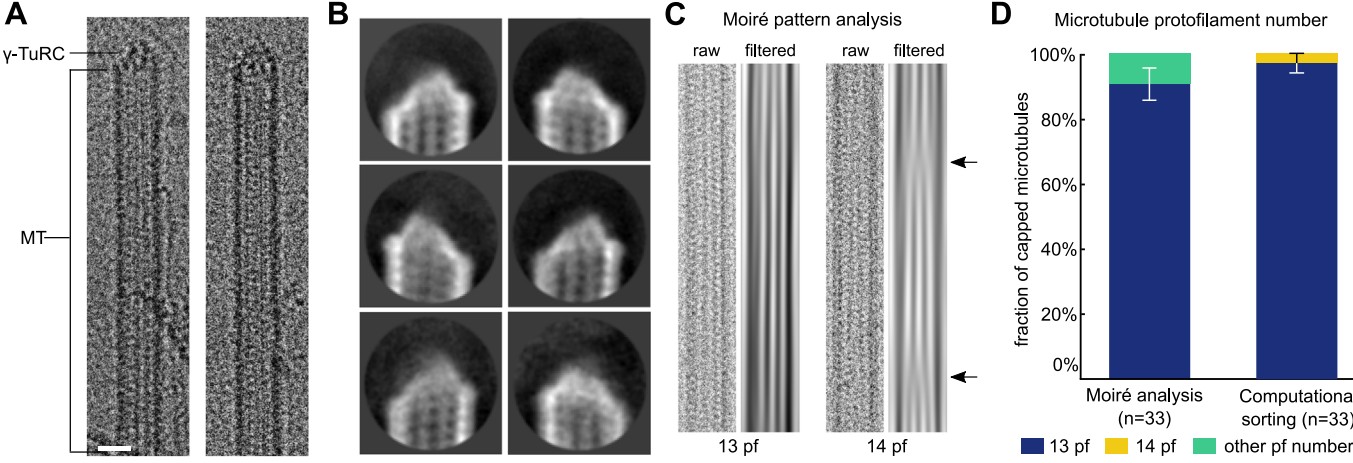

**Figure 1. Cryo-EM characterisation of γ-TuRC-capped MT minus ends purified from *X. laevis* egg extract.**

(A) Examples of γ-TuRC-capped MT minus ends identified on cryo-EM micrographs. Scale bar represents 20 nm. (B) Representative 2D classes of capped MT minus ends (mask diameter 38 nm). (C) Example raw micrograph cut-out (labelled as raw) and corresponding Moiré pattern (labelled as filtered) for a 13 protofilament (pf) and 14 protofilament MT (obtained from a non-capped MT). Arrows indicate Moiré pattern transitions characteristic of 14 protofilament MTs. (D) Fraction of Paclitaxel-stabilised, γ-TuRC-capped MTs with a specific protofilament number, as determined by Moiré pattern analysis and computational particle sorting on a subset of capped MT minus ends from n = 33 individual MTs. Error bars indicate standard deviation centred around the fraction of MTs. Source data are available online for this figure.

depicting no sign of bias from either the reference or the supplemented particles (Appendix Table S3). To the level of detail resolved, the conformation of the γ-TuRC was indistinguishable between cryo-EM reconstructions obtained with either Paclitaxel or DTX (Fig. EV3A–E). Similarly, further expansion of refinement to include the attached MT revealed the same protofilament organisation with both compounds (Fig. EV3F,G). We thus combined both datasets and obtained a cryo-EM density of the MT-capping γ-TuRC at an improved resolution of 17 Å, which was used for further analysis (Fig. 2A).

## The γ-TuRC adopts a partially closed conformation at the minus end of MTs nucleated through the RanGTP pathway

The cryo-EM reconstruction of the native MT-capping γ-TuRC purified from *X. laevis* egg extract was at sufficient resolution for

spoke-wise rigid-body docking of atomic models (Fig. 2B, Methods) and thus to discern conformational rearrangements at the subunit level. Spokes 1 to 10 of the MT-capping γ-TuRC assume a conformation reminiscent of the isolated γ-TuRC. In contrast, the last four spokes display an increasing deviation, witnessed by a pronounced lateral movement of the γ-tubulin molecules around and towards the helical axis, approaching—but not reaching—MT-compatible geometry (Fig. 2C, D; Appendix Fig. S4A–E). The separation between γ-tubulin molecules in spokes 1 and 14, measured in the plane normal to the helical axis, is reduced from 4.7 nm to 2.5 nm when comparing the isolated γ-TuRC with the MT-capping γ-TuRC (Fig. 2C). In case of perfect helicity, i.e., a closed, perfectly MT-compatible γ-TuRC, the separation between these γ-tubulin molecules would be zero, since spokes 1 and 14 would completely overlap along the helical axis. Consistent with incomplete conformational closure of the γ-TuRC, spokes 11 to 14

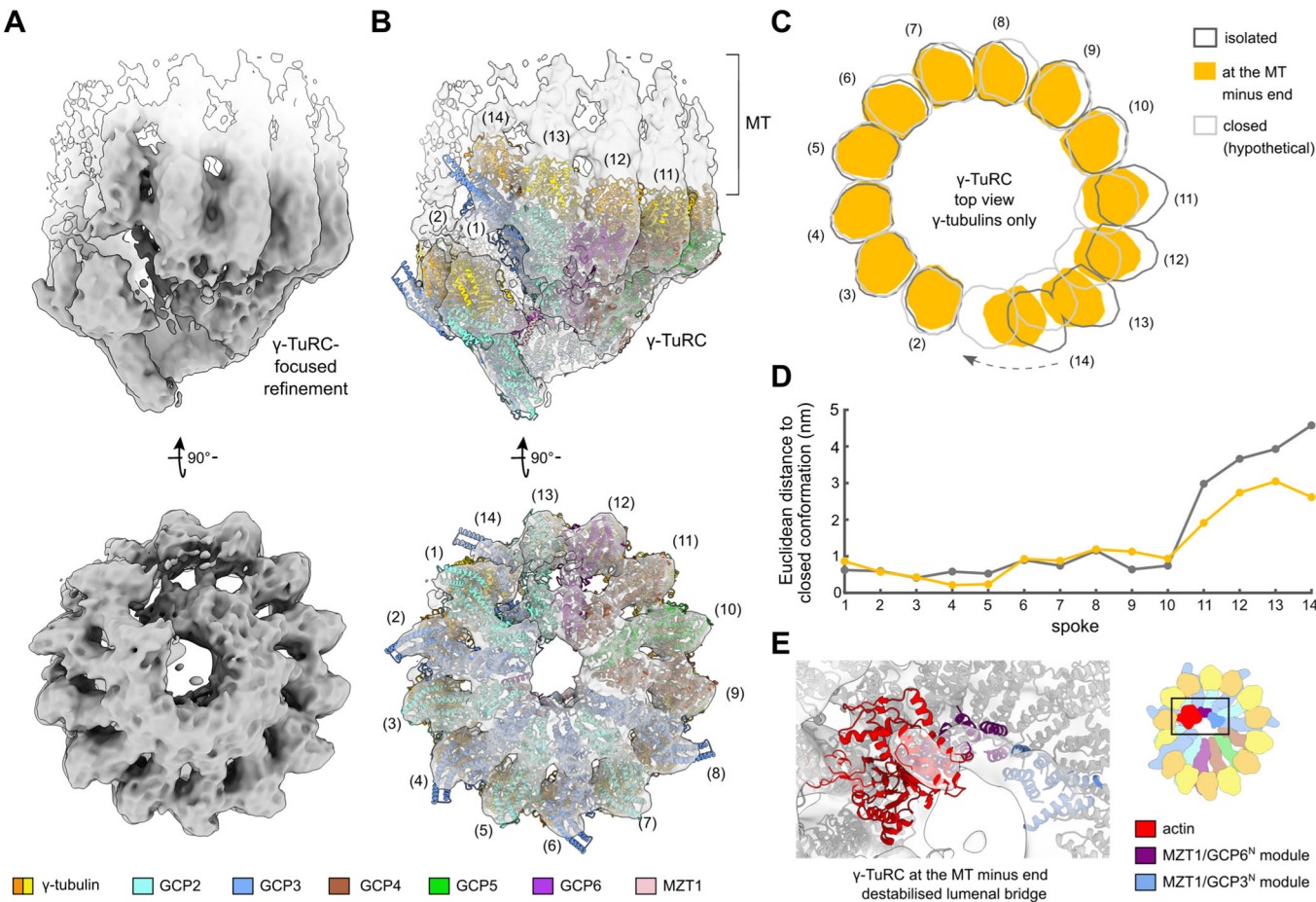

**Figure 2.  The γ-TuRC assumes a partially closed conformation at the minus end of MTs nucleated through the native RanGTP pathway.**

(**A**) Cryo-EM reconstruction of the γ-TuRC at the MT minus end. Density for the MT is shown in white. (**B**) Spoke-wise rigid-body fit of the atomic model for the *Xenopus laevis* γ-TuRC (PDB 6TF9) (Liu et al, 2020). Colouring scheme and spoke numbers are indicated. (**C**) Comparison of the γ-TuRC at the MT minus end with the isolated γ-TuRC (PDB 6TF9) (Liu et al, 2020) and the hypothetical fully closed γ-TuRC, generated based on (EMD 2799) (Kollman et al, 2015). View along the longitudinal axis. Outlines of molecular surfaces are shown for γ-tubulins of spokes 2–14; conformational change from the open to the hypothetical closed γ-TuRC at spoke 14 is indicated by an arrow. (**D**) Euclidean distance separating γ-tubulin positions in the isolated γ-TuRC (dark grey) or the MT-capping γ-TuRC (orange) from their positions in the hypothetical, fully closed γ-TuRC conformation for each spoke. Distance measured between the centres of mass of γ-tubulins. Colouring as in (**C**). (**E**) Zoom-in of the lumenal bridge in the reconstruction of the MT-capping γ-TuRC, superposed with the atomic model of the isolated *X. laevis* γ-TuRC (PDB 6TF9) (Liu et al, 2020). While the MZT1/GCP6N and MZT1/GCP3N modules of the lumenal bridge are well resolved, density for actin is reduced. Colouring as indicated; components outside the lumenal bridge are shown in grey. Source data are available online for this figure.

continue to display deviations from MT-compatible geometry of up to 1.4 nm in distance to the helical axis (Appendix Fig. S4B). Thus, the γ-TuRC partially closes at the MT minus end, but it retains pronounced conformational asymmetry, deviating from MT geometry. While likely effectively lowering the critical concentration of α/β-tubulin required for the MT nucleation reaction compared to an open γ-TuRC conformation, such a partially closed γ-TuRC presumably retains cooperative oligomerisation behaviour of α/β-tubulin subunits during nucleation (see Discussion). Hence, changes in local α/β-tubulin concentration would have a greater impact on the nucleation rate and allow for more specific regulation of the process by α/β-tubulin-concentrating factors.

Notably, the actin-containing lumenal bridge of the γ-TuRC is partially destabilised. While the two MZT1 modules part of the lumenal bridge are well resolved, EM density for the actin molecule is significantly underrepresented as compared to the *X. laevis* γ-TuRC in the open conformation (Fig. 2E; Appendix Fig. S4F). This suggests that in the partially closed conformation, actin may dissociate or have increased conformational plasticity.

## Natively nucleated capped MT minus ends are asymmetric and partially misaligned with the γ-TuRC

Previous models of the γ-TuRC-associated MT minus end assumed a perfect match between 13 exposed γ-tubulin molecules and the associated protofilaments (Brilot et al, 2021; Kollman et al, 2015). However, our unexpected finding that the MT-capping γ-TuRC isolated from *X. laevis* egg extract is not in an entirely closed conformation raises the question of how the suboptimal helical parameters of the γ-tubulin arrangement are accommodated by the MT to retain lateral protofilament-protofilament interactions, while simultaneously establishing contacts with the γ-TuRC. To address these questions, we analysed the structure of the γ-TuRC-capped MT minus end in greater detail.

While γ-TuRC-derived density segments are well resolved in the reconstruction obtained after γ-TuRC-focused 3D refinement (Fig. 2A), the density representing the emanating MT is locally less well-defined (Fig. EV4A,B), indicative of structural plasticity at the interface between γ-TuRC and MT. We therefore expanded the focus of the 3D refinement from the γ-TuRC to the entire MT minus end, enabling us to resolve all individual MT protofilaments (Figs. 3A and EV1A). Consistent with Moiré pattern analysis and computational particle sorting of capped MT minus ends (Fig. 1C,D), we observed 13 well-defined MT protofilaments growing from the γ-TuRC (Fig. 3A), in spite of the imperfect superposition of γ-tubulin 1 and 14 along the helical axis (Fig. 2C; Appendix Fig. S4A,D,E). However, in contrast to the canonical view of the 13 protofilament MT lattice, the growing MT is characterised by substantial asymmetry in terms of protofilament positions and contacts with their neighbours and the γ-TuRC.

To gain more insight into the positioning and interactions of the MT protofilaments, we created a model of the MT at capped minus ends (Fig. 3B) by rigid-body docking of protofilaments into the reconstruction, followed by rotation of each protofilament to reflect known lateral inter-protofilament contacts (Debs et al, 2020) and optimisation of the fit along the longitudinal axis (see Methods). To accommodate the larger radius of the only partially closed γ-TuRC at spokes 11 to 14 (Appendix Fig. S4B), the protofilaments associated with spokes 11, 12 and 13 are positioned further away

from the MT axis (Fig. 3C). As a consequence of the increased circumference of the MT, not all MT protofilaments engage in lateral interactions with their respective neighbours. Most strikingly, the protofilament at spoke 2 exhibits substantial gaps to both adjacent protofilaments (Fig. 3D; Appendix Fig. S5A), while at the same time a lateral misalignment of nearly 2 nm with γ-tubulin of spoke 2 exists (Fig. 3D,E). This protofilament is contacted by a density that potentially corresponds to the MZT1/GCP3$^N$ module observed on spoke 14 in the isolated γ-TuRC (Wieczorek et al, 2020a) and may contribute to stabilisation of this protofilament (Appendix Fig. S5C,D). Similarly, the protofilaments corresponding to spokes 3 to 6 are offset around 1 nm from the respective γ-tubulins when looking along the MT axis (Fig. 3C,E). This suggests weaker attachment of the MT to the γ-TuRC on the 'yeast-like' side of the ring encompassing only GCP2-GCP3 units, consistent with the less well-defined density of the corresponding protofilaments in the cryo-EM reconstruction obtained after γ-TuRC-focused 3D refinement (Fig. EV4A,B). The misalignment between MT minus ends and the partially closed γ-TuRC likely results in weaker interaction between the γ-TuRC and the associated MT compared to the interaction between layers of α/β-tubulin dimers in the MT lattice.

To rule out that shearing forces originating from MT shortening during sample purification contributed to the misalignment and asymmetry of the γ-TuRC and its associated MT, we repeated cryo-EM analysis on γ-TuRC-capped MTs that were isolated without MT shortening. The increased length of γ-TuRC-capped MTs posed challenges in terms of sample purification as well as particle distribution and density on the cryo-EM grids. Following the same approaches to cryo-EM data processing and structural analysis as above, we observed γ-TuRC and MT asymmetry indistinguishable from γ-TuRC-capped MTs to which the shortening procedure was applied (Fig. EV5A–G; Appendix Table S4, Appendix Fig. S6), confirming that MT shortening had no observable effect on the MT minus end structure.

Altogether, our data show that the geometry of the MT minus end partially adapts to the asymmetry of the partially closed γ-TuRC by an asymmetric increase in protofilament distance to the helical axis, paired with breaks in inter-protofilament and γ-tubulin-protofilament contacts (Fig. 3C,D).

## CAMSAP2 recognises the asymmetric minus end of γ-TuRC-capped MTs nucleated in *X. laevis* egg extract

Next, we hypothesised that the unique asymmetric structure of the γ-TuRC-capped MT minus end differing from the regular MT lattice may serve to recruit MT-modulating factors directly to the MT minus end. As CAMSAP2 was previously observed to bind the minus end of MTs freshly nucleated at the centrosome (Jiang et al, 2014), we aimed at probing the binding of CAMSAP2 to purified MT minus ends using multi-colour fluorescence microscopy. We incubated γ-TuRC-capped MT minus ends purified from *X. laevis* egg extracts with CAMSAP2 and analysed the CAMPSAP2 distribution pattern. CAMSAP2 displayed a strong preference for the γ-TuRC-capped MT minus end as compared to the MT lattice further away from the γ-TuRC (Fig. 4A–C; Appendix Fig. S7A–C). Consistently, CAMSAP2 can be localised at the minus end of γ-TuRC-capped MTs purified from *X. laevis* egg extracts using immunogold labelling negative stain electron microscopy

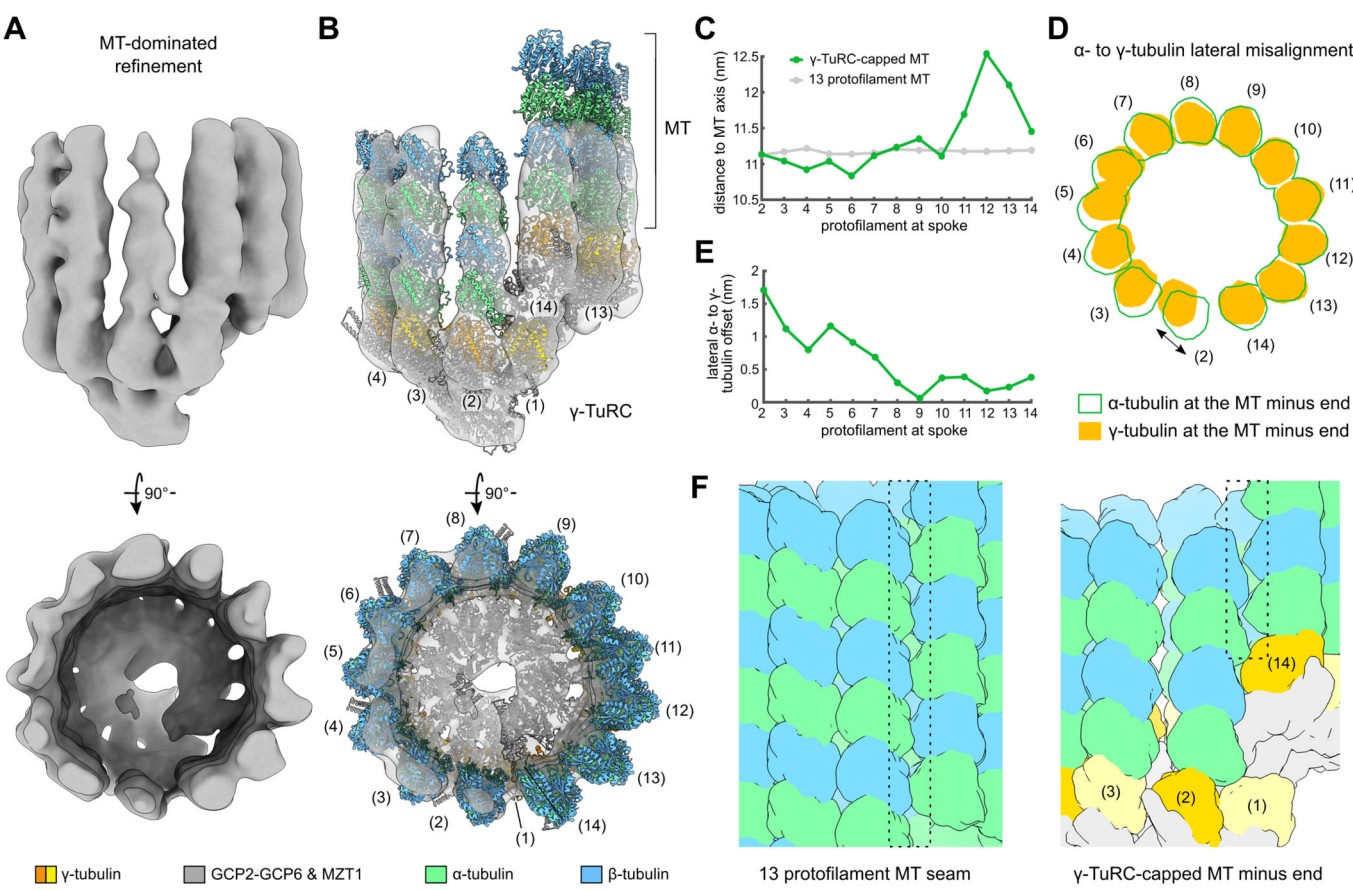

**Figure 3. Capped MT minus ends purified from *X. laevis* egg extract are asymmetric and feature breaks in inter-protofilament and protofilament-γ-tubulin contacts.**

(A) Cryo-EM reconstruction of the γ-TuRC-capped MT minus end, resulting from MT-dominated refinement. (B) Atomic models of a protofilament-wise fitted MT and the partially closed γ-TuRC (Fig. 2B) displayed in the density. Numbering is shown for γ-TuRC spokes; colour scheme is indicated at the bottom. (C) Distance from the MT axis for each protofilament for the MT at capped minus ends (green) or an ideal 13 protofilament MT (grey). (D) Molecular surfaces are shown for γ-tubulins (filled) and the first α-tubulin (outlined) of each protofilament. View along the MT axis; γ-tubulin of spoke 1 was omitted for clarity. (E) Lateral offsets (perpendicular to the MT axis) between γ-tubulin and the first α-tubulin of the respective protofilament. Parameters in (C) and (E) are measured from the respective subunit's centre of mass. (F) Left: Surface representation of the MT seam in a typical 13 protofilament MT, where lateral contacts form between α- and β-tubulin (PDB 6EW0 (Manka and Moores, 2018)). Right: Surface representation of the seam location in the minus end of capped MTs. Dashed boxes indicate the location of the seam (left) or where the seam will form (right) further along the length of the MT or after potential release from the γ-TuRC. Colouring scheme as in panel (B). Source data are available online for this figure.

(Appendix Fig. S7D–F). A similar localisation pattern could also be observed for γ-TuRC-capped MT minus ends nucleated in vitro by recombinant human γ-TuRC (Appendix Fig. S8A–I).

As CAMSAP2 is a MT-binding protein, selective recognition of γ-TuRC-capped MT minus ends via the asymmetry and deformed protofilament organisation of the emanating MT is a likely scenario, but other mechanisms, such as recognition of alternative MT features or direct binding to the γ-TuRC cannot be excluded. Altogether, the structure of the MT minus end with the associated γ-TuRC likely has direct functional consequences in promoting site-specific recruitment of MT-binding proteins relevant for processing of newly nucleated MTs, including CAMSAP proteins and potentially the CAMSAP-katanin complex (Jiang et al, 2014).

### The γ-TuRC guides the position of the MT seam

13 protofilament MTs adopt a helical configuration in which all lateral interfaces consist of homotypic α-tubulin-α-tubulin and β-tubulin-β-

tubulin contacts, except for the MT seam, where heterotypic α-tubulin-β-tubulin and β-tubulin-α-tubulin contacts are formed (Kikkawa et al, 1994; Mandelkow et al, 1986) (Fig. 3F). Based on MT minus end models that assume a closed γ-TuRC with a perfect 13 protofilament MT lattice, it has previously been suggested that the MT seam is located between γ-TuRC spokes 2 and 14 (Brilot et al, 2021; Kollman et al, 2015). However, such a model is no longer self-evident in the case of an asymmetric capped minus end, in particular due to a helical pitch lower than expected for ideal 13 protofilament MTs (Appendix Fig. S5B). For the protofilaments contacting spokes 3–14, we only observe homotypic lateral interactions, whereas the protofilament at spoke 2 does not directly contact either of the neighbouring protofilaments at the very minus end. This indicates that the seam will form between the protofilament at spoke 2 and either of its neighbours (i.e., the protofilament at spoke 3 or 14). Alongside a 1.1 nm relative upward movement of the protofilament at spoke 14, which is required for both possible seams, the protofilament at spoke 2 requires a considerably smaller relative movement to form a seam towards spoke

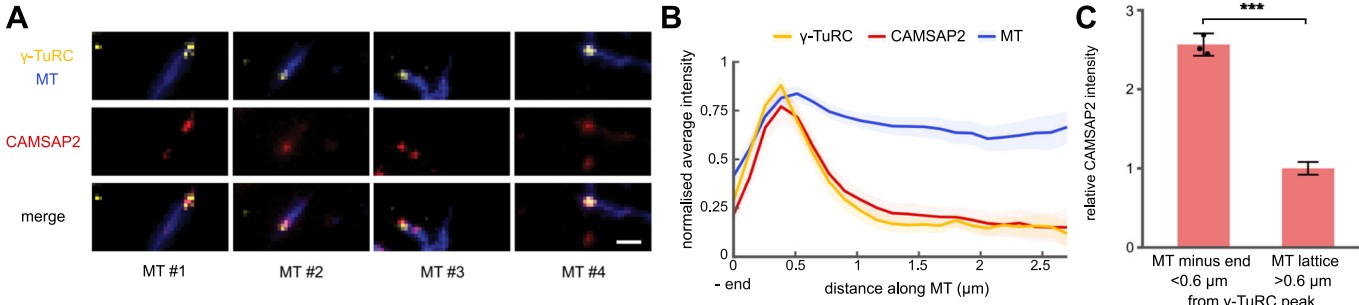

**Figure 4. CAMSAP2 specifically recognises the minus end of γ-TuRC-capped MTs nucleated through the RanGTP pathway.**

(A) Exemplary MTs nucleated through the RanGTP pathway in *X. laevis* egg extract that show colocalisation of the γ-TuRC (yellow) and GFP-CAMSAP2 (red) at the minus end of MTs (blue), as judged by fluorescence microscopy. Scale bar represents 1 μm. (B) Normalised average intensity of CAMSAP2 (red), γ-TuRC (yellow) and MT signals (blue) along the length of γ-TuRC-capped MTs ($n = 69$). Only MTs longer than 5 μm were considered; data were combined from three biological replicates. (C) Average intensity of CAMSAP2 signal within 0.6 μm of the γ-TuRC signal peak ($n = 552$ from 69 MTs, of which 28, 19 and 22 for each respective replicate) or further away ($n = 823$ from 69 MTs). Intensities were normalised to the mean CAMSAP2 intensity on the MT lattice for each individual replicate. $p = 3.2 \times 10^{-68}$ Dots indicate mean values of the three biological replicates. ***$p < 0.001$ by a one-tailed Welch's t-test. Data in (B) and (C) shown as mean with 95% confidence interval for all individual data points. Source data are available online for this figure.

14 (0.4 nm) as compared to seam formation towards spoke 3 (3.6 nm) (Fig. 3F). Notably, our cryo-EM reconstruction of the γ-TuRC-capped MT resolves a regular MT lattice at >100 nm away from the minus end (Appendix Fig. S3A–C), indicating that the seam has formed along the length of the MT through small movements between protofilaments. Altogether, our results suggest that the MT seam forms at either side of the protofilament at spoke 2 and thereby, the orientation of the γ-TuRC template can pre-determine the position of the MT seam, even before MT nucleation starts. The asymmetric order of GCP subunits within the γ-TuRC is ideally suited for positioning the γ-TuRC in a defined spatial orientation with respect to its cellular surroundings to predefine the seam.

# Discussion

Recent cryo-EM reconstructions have revealed the subunit architecture of the isolated γ-TuRC, which adopts an asymmetric, open conformation (Consolati et al, 2020; Liu et al, 2020; Wieczorek et al, 2020b), and single molecule measurements have established its MT nucleation kinetics in the presence of interaction partners (Consolati et al, 2020; Thawani et al, 2020). Nevertheless, the lack of structural information on the γ-TuRC bound to the MT minus end severely limits our mechanistic understanding of MT nucleation, as well as modulation of subsequent anchoring or release of the MT from the γ-TuRC. Here, we purified MT minus ends capped by the γ-TuRC nucleated through the native RanGTP pathway and analysed the complex using cryo-EM.

## Challenges and opportunities of the native *X. laevis* egg extract system

Compared to in vitro systems for MT nucleation, *Xenopus* egg extract offers the prime advantage of a native MT nucleation system that contains all regulatory proteins to ensure proper MT nucleation and correct attachment to the γ-TuRC, and it does not require the addition of modified proteins. However, the abundance of γ-TuRC-capped MTs in the *Xenopus* system was relatively low

and their purification at sufficient concentration for cryo-EM was challenging. As a result, particle numbers for cryo-EM analysis were limited and we had to adapt the cryo-EM data processing workflow to achieve accurate alignment of the particles. We supplemented our MT-capping γ-TuRCs with particle images of isolated γ-TuRCs, which assisted in accurate particle alignment on individual spokes and thereby enhanced the resolution of the reconstruction. Importantly, we have validated that the structure and conformation of supplemented particles did not influence the density observed for MT-capping γ-TuRCs in a series of *in silico* experiments. In the first approach, density for the GRIP2 domains and γ-tubulin molecules in spokes 5 and 6 was subtracted from the supplemented particles and the initial reference before particle refinement (Fig. EV1A,C). While these segments of spokes 5 and 6 are missing in the reconstruction of supplemented particles after joint refinement, as expected, they are clearly resolved and well covered in the reconstruction of the MT-capping γ-TuRCs, demonstrating that particles were correctly aligned without structural bias towards the supplemented particles or reference. In the second approach, we performed a series of cross-correlation experiments, systematically comparing cryo-EM reconstructions of the MT-capping γ-TuRC obtained under varying particle refinement conditions with models of the 'open', the hypothetical 'closed' and the 'partially closed' γ-TuRC (Table EV1; Appendix Tables S2–4). These experiments established that the overall conformation in the reconstruction of the MT-capping γ-TuRC is not influenced by the supplemented particles. Most importantly, refinement runs in the absence of any supplemented particles result in reconstructions of a partially closed γ-TuRC (Fig. EV2), although at lower resolution. This clearly demonstrates that the partially closed γ-TuRC conformation does not result from a bias introduced by the supplemented particles in an open conformation.

## Mechanistic implications for regulated MT nucleation

Our analysis reveals that the γ-TuRC assumes a partially closed conformation and is locally misaligned with the minus end of MTs nucleated in *X. laevis* cytoplasmic extract, which has several

mechanistic implications for MT nucleation and subsequent stable MT anchoring or release from the γ-TuRC (Fig. 5). Incomplete γ-TuRC closure may have functional advantages for the spatiotemporal regulation of the process via the α/β-tubulin concentration. For a fully closed γ-TuRC, the MT nucleation rate is expected to increase linearly with α/β-tubulin concentration (Thawani et al, 2020). In contrast, a partially closed γ-TuRC is likely to retain cooperative binding of α/β-tubulin dimers, while still effectively lowering the critical concentration of α/β-tubulin compared to nucleation by an open γ-TuRC (Fig. 5A). Changes in α/β-tubulin concentration would thus have a greater impact on the nucleation rate and allow for more specific regulation of the process. Local increase of α/β-tubulin concentration can be achieved through co-condensation with a number of factors (Baumgart et al, 2019; Duan et al, 2023; Hernandez-Vega et al, 2017; Imasaki et al, 2022; Jiang et al, 2015; Montenegro Gouveia et al, 2018; Niedzialkowska et al, 2024; Sun et al, 2021; Trivedi et al, 2019; Woodruff et al, 2017), including TPX2 (King and Petry, 2020), or through the local release of α/β-tubulin from tubulin-sequestering factors such as stathmin (Niethammer et al, 2004). In addition, γ-TuRC-interacting MT polymerases from the XMAP215/chTOG/Stu2 family effectively increase α/β-tubulin concentration directly at the γ-TuRC (Gunzelmann et al, 2018; Thawani et al, 2020). Such a non-linear, multifactorial activation mechanism (Stobe et al, 2009) increases the spatial and temporal specificity of MT nucleation.

The misalignment between the first five γ-TuRC spokes and the accompanying MT protofilaments (Fig. 3D,E) further raises the possibility of a point nucleation mechanism, in which the initial steps of nucleation always take place on the γ-TuRC spokes where lateral misalignment does not occur, i.e., on the second half of the ring (spokes 6–14). Such a nucleation mechanism, relying only on specific segments of the ring for the initial steps, would explain why actin binding-deficient γ-TuRCs with altered geometry (Wurtz et al, 2022) or even γ-TuRC subcomplexes resembling half rings (Wieczorek et al, 2021) do not display significantly diminished nucleation activity in vitro. Still, a structurally intact γ-TuRC is likely required for regulation of MT nucleation and anchoring in cells, as indicated by the compromised MT nucleation kinetics of actin binding-deficient γ-TuRCs in vivo (Wurtz et al, 2022). Indeed, we observe a structural reorganisation at the actin-containing lumenal bridge after MT nucleation through a native pathway, indicating it may have a role during the regulation of MT nucleation and recycling of γ-TuRCs. Presently, it is unclear whether structural modulation of the lumenal bridge is an indirect effect of γ-TuRC conformational changes during the MT nucleation reaction, or whether direct binding or modification of lumenal bridge components, for example by kinases, actively contribute to the process. In light of this finding, it will be important to study actin regulation and turnover within the γ-TuRC during the MT nucleation reaction in further detail.

## Mechanistic implications for MT minus end modulation

Our observation that CAMSAP2 can specifically bind to the γ-TuRC-capped MT minus end suggests that its architecture creates a specific binding site for minus end modulating proteins distinct from the regular MT lattice further along the MT (Fig. 5B). Consistently, Rai et al (2024) recently demonstrated that the other two CAMSAP family members, CAMSAP1 and CAMSAP3,

similarly bind γ-TuRC-capped MT minus ends (Rai et al, 2024). In addition to potentially providing a specific binding site for minus end-modulating proteins, misalignment between the MT minus end and the partially closed γ-TuRC likely substantially lowers the affinity of γ-TuRC for the associated MT and thereby predefines a breaking point for MT release (Fig. 5B), as observed for in vitro nucleated MTs through binding of CAMSAP2/3 (Rai et al, 2024) or the concerted action of the depolymerising kinesin KIF2A and the MT severing enzyme spastin (Henkin et al, 2023). Such controlled and specific release of the MT from the γ-TuRC facilitates the liberation of nucleation-competent γ-TuRC, which would be particularly beneficial in processes where generation and turnover of large numbers of MTs is required, like during spindle formation. Simultaneously, this would be consistent with observed dynamic MT behaviour at the minus end and thereby enable MT treadmilling (Henkin et al, 2023; Waterman-Storer and Salmon, 1997), i.e., elongation at the plus end while the minus end depolymerises.

While we demonstrate that the γ-TuRC can adopt a partially closed state with a misaligned MT minus end after RanGTP-induced nucleation, we cannot exclude that the γ-TuRC adopts other states in different situations, e.g. in the context of the centrosome, where CDK5RAP2 promotes MT nucleation (Choi et al, 2010; Muroyama et al, 2016; Rale et al, 2022) (Fig. 5C). CDK5RAP2 suppresses CAMSAP binding and subsequent release of the MT from γ-TuRC-capped minus ends in vitro (Rai et al, 2024). One possible explanation for this observation is that CDK5RAP2 induces closure of the γ-TuRC to MT symmetry, thereby eliminating the misalignment between the MT and its γ-TuRC cap. Similarly, the γ-TuRC may fully close at different phases during the nucleation process to favour a high MT nucleation rate and subsequently relax to a partially closed conformation that favours MT release (Fig. 5D). Such a mechanism of conformational relaxation could be mediated by GTP hydrolysis in the MT or the γ-TuRC itself or by a transient binding partner that leaves the γ-TuRC after MT nucleation.

Indeed, recent work released during the revision of this manuscript described cryo-EM structures of closed γ-TuRCs with symmetric 13-protofilament MTs attached (preprint: Aher et al, 2023; preprint: Barford et al, 2023; Brito et al, 2024). One of these studies (preprint: Barford et al, 2023) used the evolutionarily strongly simplified budding yeast model system only encoding for three γ-TuRC components (GCP2, GCP3 and γ-tubulin), in which the γ-TuRC is inherently symmetric and samples a fully closed structure already before MT nucleation (Brilot et al, 2021). Two other studies (preprint: Aher et al, 2023; Brito et al, 2024) demonstrated that the vertebrate γ-TuRC can adopt a fully closed conformation bound to a symmetric MT when nucleated in vitro. The in vitro systems included γ-TuRC with undefined post-translational modification state (from recombinant source or unsynchronised cells), mutant α-tubulin to attenuate GTP hydrolysis kinetics (Roostalu et al, 2020) and only one specific β-tubulin isotype (TUBB3) with non-canonical oligomerisation properties (Ti et al, 2018). Moreover, the in vitro systems did not include a full physiological set of regulatory proteins for γ-TuRC-based microtubule nucleation. Thus, the composition of the in vitro microtubule nucleation systems may have converged into a MT nucleation pathway different from the native RanGTP-induced pathway we studied, resulting in a fully closed structure of the

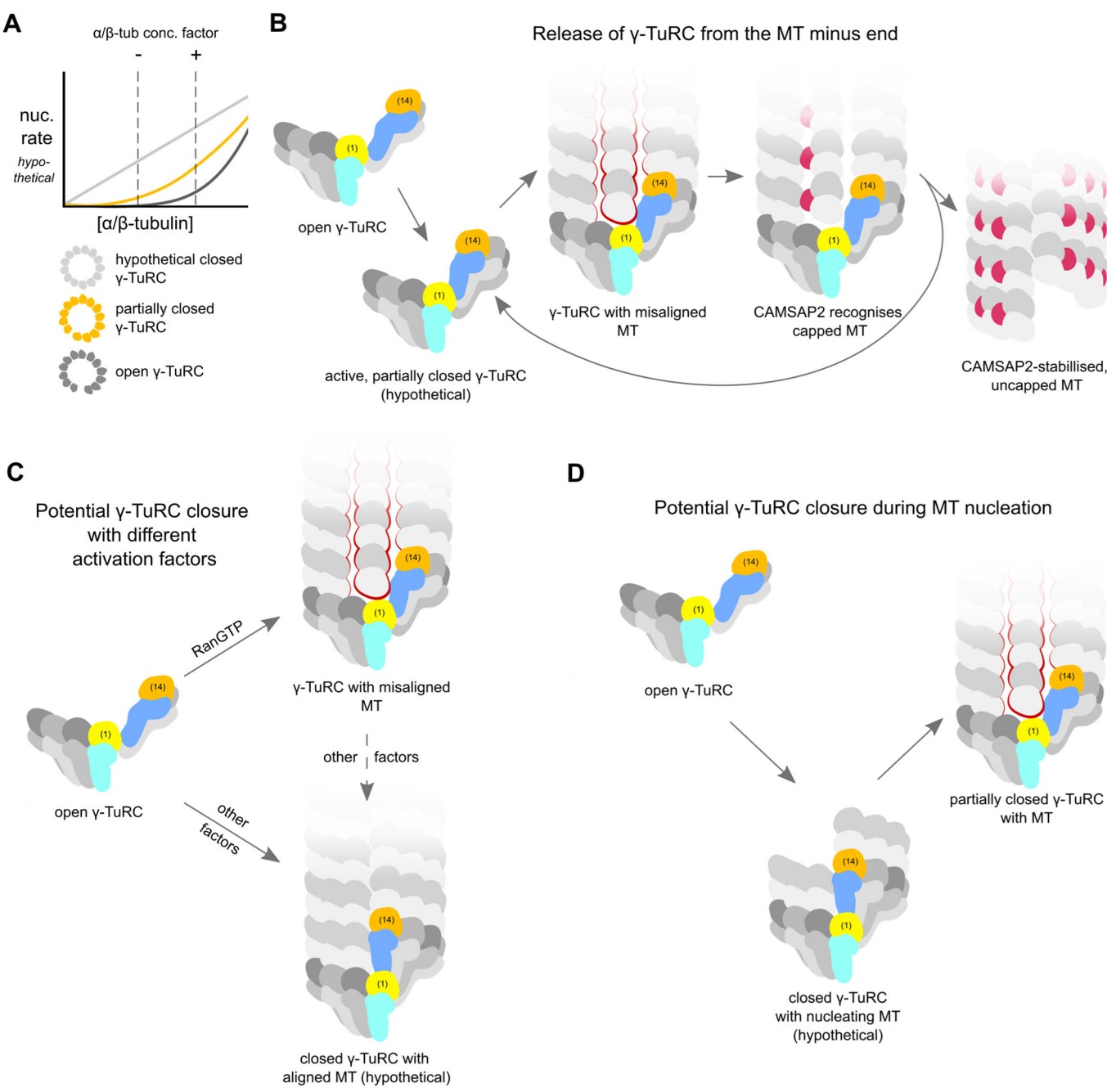

**Figure 5.  Possible roles for partial γ-TuRC closure and MT misalignment in the regulation of MT nucleation and release.**

(A) The open γ-TuRC (dark grey curve) displays highly cooperative nucleation behaviour in response to α/β-tubulin concentration but has a low base rate of nucleation (Thawani et al, 2020). The fully closed γ-TuRC (light grey) has a high base rate of nucleation with a lack of cooperativity. A partially closed γ-TuRC (orange) would likely have an intermediate base rate of nucleation, while still retaining some cooperativity. Such behaviour could aid synergy of the γ-TuRC with α/β-tubulin-enriching factors. Graphs are for illustrative purposes and do not represent experimental or simulated data. (B) Misalignment between the γ-TuRC and MT protofilaments may promote release of freshly nucleated MTs from the γ-TuRC, e.g. by CAMSAP2 (red), and recycling of nucleation-competent γ-TuRC. Spoke 1 and 14 are coloured to highlight different degrees of γ-TuRC closure. Spoke numbers are indicated for spokes 1 and 14. Missing interactions around the protofilament at spoke 2 highlighted with red outlines. (C) Different activation cues may induce different degrees of γ-TuRC closure to accommodate differential requirements of nucleation rate and minus end stability. (D) The γ-TuRC may transiently visit a fully closed conformation that favours MT nucleation and subsequently relax to the partially closed conformation observed in our study to favour MT release.

minus end-capping γ-TuRC. Notably, Brito et al (2024) also observed MT-bound γ-TuRC in a partially closed conformation, indicating that the γ-TuRC can assume open, partially closed and fully closed conformations to control MT nucleation rate and MT affinity of the γ-TuRC. The partially closed conformation observed here may thus serve the specific demands of the dynamic RanGTP/branching MT nucleation pathway, in which thousands of microtubules are formed within a short period of time. In this scenario, a mismatch between the γ-TuRC and nucleated microtubules may serve as a defined breaking point that promotes fast turnover of γ-TuRCs.

# Methods

## Research animals

Husbandry of *Xenopus laevis* female frogs (Xenopus1, Dexter, MI, USA) was performed in a XB60 Stand-alone Xenopus Rack facility (Aqua Schwarz, Göttingen, Germany). Husbandry was approved by the responsible authority, i.e. the City of Bonn, quoting §11 Abs. 1, Nr.1 of the German law for animal protection under file reference 76-5/2022/, Amt für Umwelt und Stadtgrün according to EU guideline 2010/63 for aquatic anura. Animal experiments (hormone applications for egg production) were approved by the respective authority, i.e. Landesamt für Natur, Umwelt und Verbraucherschutz Nordrhein-Westfalen, under the file reference 81-02.04.40.2022.VG027.

In brief, animals were kept at a constant temperature of 18 °C in water with salt supplement reaching to 1200 µS, pH 7.0, with an automatic 12 h daily light-dark-cycle. Feeding (SDS Aquatic 3, LBS, Techniplast, Gams, Switzerland) was performed twice weekly for 15 min. Hormones (human HCG, Intervet, Unterschleißheim, Germany) were applied in a first dose using 75 U (priming) four days before the second dose (500 U), animals held further for 16 h in individual boxes in MMR, eggs collected and egg extracts (CSF arrested) prepared as described previously (Maresca and Heald, 2006).

*Xenopus* frogs were held solely for the purpose of egg production, i.e., the physiological process of spating, and no further animal or in vivo experiments were performed. ARRIVE 2.0 were carefully checked but not found applicable for the experiments performed with cell-free extracts.

## Expression and purification of Ran(Q69L)

Ran(Q69L), cloned into a pQE32 vector, was transformed into *Escherichia coli* competent cells (strain BL21-CodonPlus(DE3)-RIL). Expression was induced with 0.2 mM IPTG (Roth) at an optical density of 0.5–0.8 in 2xYT medium. Cells were grown at 30 °C for 16 h. Afterwards, cells were collected, washed with PBS and cell pellet was stored at −80 °C. The pellet was resuspended in lysis buffer (10 mM HEPES pH 7.6, 100 mM KCl, 1 mM $MgCl_2$, 5% v/v glycerol, 1 mM DTT, 0.1% Triton X-100, cOmplete EDTA-free protease inhibitor cocktail (Roche) and PMSF), lysed by sonication and clarified by centrifugation at 40,000 rpm for 20 min in a Type 50.2 Ti Rotor (Beckman Coulter). Supernatant was incubated with Ni-NTA Agarose beads (Qiagen) for 2 h at 4 °C, washed with lysis buffer and eluted with lysis buffer supplemented with 300 mM

imidazole (Roth). Imidazole was removed via buffer exchange through a HiTrap Desalting column (5 mL, Cytiva) on an ÄKTA go system (Cytiva) operated using the UNICORN software (version 7.6). Ran(Q69L) was concentrated to 350 µM and stored at −80 °C.

## Purification of γ-TuRC-capped microtubule minus ends from *X. laevis* egg extracts

Cytostatic factor (CSF)-arrested meiotic egg extracts from genetically unmodified, mature (i.e. more than 9 months old) female *Xenopus laevis* (lab bread NASCO, USA) were prepared according to Chinen et al (Chinen et al, 2015) Low-speed supernatant was obtained by centrifugation at $100,000 \times g$ for 60 min at 4 °C in a S120-AT2 rotor (Sorvall). *Xenopus* egg extract quality was verified on a small aliquot by its aster-forming capacity after addition of 20 µM Ran(Q69L) and 4 µM Cy3-labelled pig brain tubulin and incubation at 20 °C for 20 min, after which 3 µL of sample was squash-fixed on a slide with a $12 \times 12$ mm coverslip and analysed by fluorescence microscopy (Nikon Ti-E epifluorescence microscope with a 10× objective (0.25 NA, Nikon), a $2048 \times 2048$ px (6.5 µm) sCMOS camera (Flash4, Hamamatsu) and a Perfect Focus autofocus system (Nikon), using 542/27 excitation and 600/52 emission filters (Semrock)). If extract was of satisfactory quality, low-speed supernatant was mixed with 20 µM of GTPase-deficient Ran(Q69L) and incubated for 20 min in a 20 °C water bath to induce MT aster formation, and then mixed with 20 µM Paclitaxel (Sigma-Aldrich) or docetaxel (Sigma-Aldrich) and incubated for an additional 10 min in a 20 °C water bath to stabilise the nucleated MTs. Formed MT asters were separated via centrifugation through a 40% glycerol cushion ((1xBRB80 (80 mM PIPES/KOH pH 6.8, 1 mM $MgCl_2$ and 1 mM EGTA), 40% glycerol, 10 µM Paclitaxel or docetaxel, 0.1% Triton X-100)) at $53,000 \times g$ for 20 min at 20 °C in a S120-AT2 rotor (Sorvall). Pellet was resuspended in BRB80 buffer (80 mM PIPES/KOH pH 6.8, 1 mM $MgCl_2$ and 1 mM EGTA) supplemented with 1 mM GTP and 10 µM Paclitaxel or docetaxel. To prevent overcrowding of cryo-EM grids with long MTs of which the largest part is distant from their associated γ-TuRC, MTs were fragmented prior to affinity purification inspired by a previously published protocol, with several modifications (Collins and Vallee, 1987). MTs were incubated for 5 min on ice in the presence of 3 mM $CaCl_2$, during which extensive pipetting was applied. Then, 100 mM NaCl was added to the MTs. For γ-TuRC-capped MTs to which no shortening procedure was applied, these fragmentation steps were omitted. To separate γ-TuRC-capped MTs from uncapped MTs, the sample was incubated with anti-γ-tubulin antibody coupled to Dynabeads Protein A (Life Technologies) for 30 min at room temperature, followed by three rounds of washing with BRB80 supplemented with 100 mM NaCl, 1 mM GTP and 10 µM Paclitaxel or docetaxel. γ-TuRC-capped MTs were eluted for 2 h at room temperature in BRB80 containing 0.1 mg/ml γ-tubulin antigenic C-terminal peptide (Zheng et al, 1995), 1 mM GTP, 10 µM Paclitaxel or docetaxel and 0.02% Tween 20. Sample was used directly.

## Mass spectrometry sample preparation

Sample of γ-TuRC-capped MTs isolated from *X. laevis* egg extract with Paclitaxel stabilisation was prepared using the SP3 protocol (Mikulasek et al, 2021) and trypsin (sequencing grade, Promega)

was added in an enzyme to protein ratio of 1:50 and incubated for 4 h at 37 °C in 50 mM HEPES supplemented with 5 mM TCEP and 20 mM CAA. For further sample clean up, an OASIS® HLB µElution Plate (Waters) was used according to manufacturer's instructions.

## LC-MS/MS

An UltiMate 3000 RSLC nano LC system (Dionex) fitted with a trapping cartridge (µ-Precolumn C18 PepMap 100, 5 µm, 300 µm i.d. × 5 mm, 100 Å) and an analytical column (nanoEase™ M/Z HSS T3 column 75 µm × 250 mm C18, 1.8 µm, 100 Å, Waters) was used. Trapping was carried out with a constant flow of trapping solution (0.05% trifluoroacetic acid in water) at 30 µL/min onto the trapping column for 6 min. Subsequently, peptides were eluted via the analytical column running solvent A (3% DMSO, 0.1% formic acid in water) with a constant flow of 0.3 µL/min, with increasing percentage of solvent B (3% DMSO, 0.1% formic acid in acetonitrile). The outlet of the analytical column was coupled directly to an Orbitrap Qexactive™ plus Mass Spectrometer (Thermo) using the Nanospray Flex™ ion source in positive ion mode.

The peptides were introduced into the Qexactive plus via a Pico-Tip Emitter 360 µm OD × 20 µm ID; 10 µm tip (CoAnn Technologies) and an applied spray voltage of 2.2 kV. The capillary temperature was set at 275 °C. Full mass scan was acquired with mass range 350–1500 $m/z$ in profile mode with resolution of 70,000. The filling time was set at maximum of 100 ms with a limitation of $3 \times 10^6$ ions. Data dependent acquisition (DDA) was performed with the resolution of the Orbitrap set to 17,500, with a fill time of 50 ms and a limitation of $1 \times 10^5$ ions. A normalised collision energy of 26 was applied. Dynamic exclusion time of 20 s was used. The peptide match algorithm was set to 'preferred' and charge exclusion 'unassigned', charge states 1, 5–8 were excluded. MS$^2$ data was acquired in profile mode.

## Mass spectrometry data analysis

All raw files were converted to mzmL format using MSConvert from Proteowizard (Chambers et al, 2012), using peak picking from the vendor algorithm. Files were then searched using MSFragger v3.7 (Kong et al, 2017) in Fragpipe v19.1 against the Swissprot *Xenopus laevis* database UP000186698 (34806 entries) containing common contaminants and reversed sequences. The standard settings of the Fragpipe LFQ workflow were used. The following modifications were included into the search parameters: Carbamidomethyl I (fixed modification); Acetyl (Protein N-term) and Oxidation (M) (variable modifications). For the full scan (MS1), a mass error tolerance of 10 ppm and for MS/MS (MS2) spectra of 0.02 Da was set. Further parameters were set: Trypsin as protease with an allowance of maximum two missed cleavages and a minimum peptide length of seven amino acids was required. The false discovery rate on peptide and protein level was set to 0.01. A full list of mass spectrometry hits is available as source data.

## Expression and purification of CAMSAP2

Human CAMSAP2 was recombinantly expressed as a fusion protein with N-terminally tagged 6His-GFP in a pFastBac vector

(pFastBac+GFP-CAMSAP2 was a gift from Ron Vale (Addgene plasmid # 59037; http://n2t.net/addgene:59037; RRID:Addgene_59037) (Hendershott and Vale, 2014)). Baculovirus was produced in Sf21 insect cells after transfection with cellfectin II reagent (Thermo Fisher Scientific). Infected Sf21 cells were incubated at 27 °C for 60 h, followed by harvesting via centrifugation (800 × $g$ for 3 min), flash freezing and storage at –80 °C until protein purification. The insect cell pellet was resuspended in lysis buffer (50 mM HEPES pH 7.6, 300 mM KCl, 1 mM MgCl$_2$, 1 mM EGTA, 1 mM DTT, 0.1% Triton X-100, cOmplete EDTA-free protease inhibitor cocktail (Roche) and PMSF), sonicated for 3 × 1 min with 0.6 amplitude and 0.5 cycles (Hielscher UP50H) and centrifuged for 30 min at 20,000 × $g$ and 4 °C. The supernatant was incubated with Ni-NTA Agarose beads (Qiagen) for 2 h at 4 °C, washed with lysis buffer and eluted with lysis buffer supplemented with 300 mM imidazole (Roth). Imidazole was removed via buffer exchange through a HiTrap Desalting column (5 mL, Cytiva) on an ÄKTA go system (Cytiva). Proteins were flash-frozen and stored at –80 °C until usage.

## Expression and purification of recombinant γ-TuRC

Recombinant human γ-TuRC was expressed and purified as described before (Wurtz et al, 2022), flash-frozen and stored at –80 °C. Briefly, bacmids encoding the recombinant γ-tubulin complex (*2xFLAG-TUBGCP5, TUBGCP6, TUBGCP4, TUBG1, ACTB, TUBGCP2, TUBGCP3, MZT1, MZT2B, RUVBL1* and *RUVBL2*) were produced using the MultiBac (GENEVA Biotech) system and baculoviruses were produced in Sf21 insect cells transfected using cellfectin II (Thermo Fisher Scientific). Infected cells were incubated at 27 °C for 60 h, harvested, flash frozen and stored at –80 °C. Cells were lysed in 15 mL lysis buffer (50 mM Tris-HCl, pH 7.5, 200 mM NaCl, 1 mM MgCl$_2$, 1 mM EGTA, 0.5 mM DTT, 0.1% (vol/vol) Tween-20 with 10 µL Benzonase (Sigma-Aldrich) and one cOmplete EDTA-free protease inhibitor tablet (Roche) by sonication (3 × 1 min with 0.6 amplitude, Hielscher UP50H) and centrifuged at 20,000 × $g$ for 30 min at 4 °C. Equilibrated anti-FLAG M2 Magnetic beads (Millipore Sigma, #M8823) were added to the supernatant and incubated for 1–2 h with rotation at 4 °C. Sample was washed once with lysis buffer and twice with wash buffer (50 mM Tris-HCl, pH 7.5, 150 mM NaCl, 1 mM MgCl$_2$, 1 mM EGTA, 0.5 mM DTT). Complex was eluted with wash buffer supplemented with 0.5 mg/ml 3x FLAG peptide (Gentaur).

## Purification of in vitro reconstituted γ-TuRC-capped microtubule minus ends

60 µM pig brain tubulin, supplemented with 4% Cy3-labelled tubulin, in BRB80 buffer (80 mM PIPES/KOH pH 6.8, 1 mM MgCl$_2$, 1 mM EGTA) with 12.5% glycerol was spun down for 5 min at 352,860 × $g$ and 4 °C in a S100-AT3 rotor (Thermo Fisher Scientific). The supernatant was mixed with recombinant human γ-TuRC (1:4 diluted γ-TuRC (5–10 nM) in BRB80 with 12.5% glycerol, 2 mM GTP) at a 1:1 ratio. Initial MT polymerisation was started for 5 min at 37 °C, after which 10 µM Paclitaxel was added and incubated for another 5 min. Polymerised MTs were twice spun down through a 30% glycerol cushion in BRB80 with 10 µM Paclitaxel and 0.1 mM GTP for 20 min at 69,700 × $g$ and 30 °C.

After resuspension, KCl was added to 100 mM. MTs were transferred to equilibrated anti-FLAG M2 magnetic beads (Millipore Sigma, #M8823), incubated for 1 h at room temperature, washed 3 times (BRB80, 100 mM KCl, 10 μM Paclitaxel, 1 mM GTP) and eluted with wash buffer supplemented with 0.2 mg/ml 3xFLAG peptide (Gentaur) and 0.02% Tween 20 for 1 h at room temperature. Purified γ-TuRC-capped microtubule minus ends were immediately used for analysis.

## Immunofluorescence and fluorescence microscopy

200 nM CAMSAP2 was mixed with γ-TuRC-capped MTs freshly purified from *X. laevis* egg extract (with the addition of 4 μM Cy3-labelled pig brain tubulin simultaneously with Ran(Q69L)) or in vitro reconstituted γ-TuRC-capped microtubule minus ends and incubated at 37 °C for 30 s. Samples were subsequently crosslinked with 0.5% glutaraldehyde. MTs with CAMSAP2 were spun onto a 12 mm coverslip through 10% glycerol cushion in BRB80 in a Corex 12 mL glass tube with glass support platform in the bottom. Centrifugation was performed at 23,530 × *g* for 1 h at 20 °C in an HB6 rotor (Thermo Fisher Scientific). After fixing with cold methanol for 5 min, the coverslip was washed with PBS and blocked with PBS supplemented with 10% FBS and 0.1% Triton X-100 for 20 min at room temperature. Afterwards, the coverslip was incubated with primary antibody (homemade rabbit anti-γ-tubulin antibody for native γ-TuRC-capped MTs; anti-γ-tubulin, ab27074, Sigma-Aldrich for in vitro reconstituted γ-TuRC-capped MTs) in 3% BSA for 40 min and subsequently with Alexa Fluor-conjugated secondary antibody (647 donkey anti-rabbit IgG (H + L), LOT 2284672 for native γ-TuRC-capped MTs; 680 donkey anti-mouse, LOT 1853988, Life technologies for in vitro reconstituted γ-TuRC-capped MTs) for 20 min at room temperature. Between incubation steps, the coverslip was washed with PBS. Samples were washed 3x with PBS and mounted with Mowiol. Data were collected on a DeltaVision RT system (Applied Precision, Olympus IX71-based) equipped with a Photometrics CoolSnap HQ camera (Roper Scientific), a 60x/1.42 NA UPlantSAPO objective (Olympus), a mercury arc light source and the softWoRx software (Applied Precision). Exposure times of FITC, TRITC and Cy5 channels were adjusted individually to the fluorescence intensities of respective proteins.

For analysis of CAMSAP2 binding specificity along the lattice of γ-TuRC-capped MTs, a segmented line profile (3 px line width, spline fit) of 69 (for native γ-TuRC-capped MTs) or 159 (for in vitro reconstituted γ-TuRC-capped MTs) γ-TuRC-capped MTs (independent of CAMSAP2 colocalisation with the γ-TuRC) was sampled every 0.13 μm (for native γ-TuRC-capped MTs) or 0.21 μm (for in vitro reconstituted γ-TuRC-capped MTs) in FIJI (Schindelin et al, 2012). Analysis was performed on three independent replicates of isolation or reconstitution of γ-TuRC-capped MT minus ends. For each MT, the average background signal was subtracted. Normalisation was performed independently for each MT (for native γ-TuRC-capped MTs) or each replicate (for in vitro reconstituted γ-TuRC-capped MTs).

## Negative stain EM

Copper-palladium 400-mesh EM grids (PLANO GmbH) with a continuous carbon layer of ~10 nm were glow-discharged (ACE1,

GaLa Instrumente GmbH), before 5 μL of sample was applied and incubated for ~30 s at room temperature. After blotting on a Whatman filter paper 50 (CAT N.1450-070), grids were rinsed three times on a drop of water. Grids were stained using three drops of 3% uranyl acetate in water.

MT images were acquired on a JEOL JEM-1400 (JEOL Ltd., Tokyo, Japan) operating at 80 kV, equipped with a 4k × 4k digital camera (F416, TVIPS, Gauting, Germany). Micrographs were adjusted in brightness and contrast using ImageJ 2.0 (Rueden et al, 2017). 427 micrographs of the recombinant human γ-TuRC preparation used for in vitro MT nucleation and CAMSAP2 binding experiments were collected on a Talos L120C TEM equipped with a 4k × 4k Ceta CMOS camera (Thermo Fisher Scientific) at a pixel size of 3.28 Å (45,000× magnification) and a nominal defocus of −2 μm.

Negative stain EM of recombinant human γ-TuRCs was processed in RELION 3.1 (Zivanov et al, 2018). After estimation of CTF parameters with Gctf (Zhang, 2016), 246,121 particles were automatically picked, extracted with a box size of 128 px at full spatial resolution. After 4 consecutive rounds of 2D classification, 2D classes representing γ-TuRCs (42,828 particles) were selected and subjected to 3D classification, using a reference density of the γ-TuRC with the GRIP2 domains and γ-tubulin molecules of spokes 5 and 6 removed as a control for bias (Wurtz et al, 2021). All well-aligning particles formed a class showing the γ-TuRC with 14 spokes, indicative of full γ-TuRC assembly (Appendix Fig. S8A–C).

## Immunogold labelling and negative stain EM

γ-TuRC-capped MT minus ends from *Xenopus laevis* egg extract were incubated with or without 200 nM GFP-CAMSAP2 for 30 s and subsequently crosslinked with 0.5% glutaraldehyde. 5 μl of sample was applied on a glow-discharged carbon-covered copper-palladium 400-mesh grid for 1 min at RT, rinsed on two drops of water and incubated for 20 min with a drop of blocking buffer (1.5% BSA in PBS supplemented with 0.1% fish skin gelatin, FSG). Grids were subsequently incubated for 30 min with mouse anti-γ-tubulin antibody (Sigma; 1:20 dilution). After rinsing with 3 drops of PBS, grids were incubated with rabbit anti-mouse linker antibody (Abcam; 1:200) for 20 min, rinsed again with PBS and incubated with Protein A gold conjugate (PAG, 15 nm gold, Utrecht University, Utrecht, The Netherlands) for 15 min. Grids were blocked again for 10 min and incubated with goat anti-GFP antibody (Rockland; 1:20) for 30 min, rinsed with 3 drops of PBS and subsequently incubated with protein A gold-conjugated rabbit anti-goat antibody (1:50; 10 nm gold) for 20 min. After thorough rinsing with PBS, sample on grids was fixed with 1% glutaraldehyde in PBS for 3 min and then thoroughly rinsed with distilled water, before continuing with post-staining with 3% uranyl acetate (UA) in water. Excess UA was blotted away with Whatman 50 filter paper and dried in air.

## Cryo-EM sample preparation

Holey carbon grids (Cu R2/1, 200 mesh; Quantifoil) were glow-discharged for 20 s in a Gatan Solarus 950 plasma cleaner. 5 μL of γ-TuRC-capped microtubule minus ends purified from *Xenopus laevis* egg extract was applied to the carbon face of the grid inside

the chamber of a Vitrobot Mark IV (Thermo Fisher/FEI), maintained at 100% humidity and 22 °C. After 30 s of waiting time, grids were blotted from both sides and immediately plunge-frozen into liquid ethane. Grids were stored in liquid nitrogen until further use.

## Cryo-EM data acquisition

Cryo-EM data were acquired using a Titan Krios (Thermo Fisher/FEI) operated at 300 kV, equipped with a Gatan K3 direct electron detector and a Quantum Gatan Imaging Filter (Gatan) with an energy slit width of 20 eV. Micrograph movies were acquired at a nominal defocus ranging between −1 μm to −3 μm, and a nominal magnification of 33,000× (2.54 Å/px) in EPU (Thermo Fisher/FEI). A cumulative electron dose of 38–47 e⁻/Å² was applied per movie, distributed over 30–60 fractions. 15,513 (Paclitaxel, 4 acquisition sessions), 17,095 (DTX, 3 acquisition sessions) or 37,279 (Paclitaxel, no shortening applied, 5 acquisition sessions) movies were recorded.

## Cryo-EM data processing

All data processing (Fig. EV1) was performed in RELION 3.1 (Zivanov et al, 2018), unless noted otherwise. Beam-induced motion was corrected with MotionCor2 (Zheng et al, 2017) using 5 × 5 patches. CTF parameters were subsequently estimated using Gctf (Zhang, 2016). For dataset 3 of the Paclitaxel sample, micrographs with an estimated maximum resolution of >8 Å were discarded. 9382 (Paclitaxel) or 5896 (DTX) candidate particles were manually picked on the remaining 14,666 (Paclitaxel) or 17,095 (DTX) micrographs and extracted at full spatial resolution. Picked particles were subjected to multiple successive rounds of 2D classification, in which capped MT minus end classes were retained, mixed classes were sent into the next round of classification and uncapped ends and capped ends too close to the lattice of neighbouring MTs were discarded. Then, the 4873 (Paclitaxel) and 3624 (DTX) remaining particles, 8497 particles in total, were subjected to 2D classification into 1 class to align all MT minus ends. Subsequently, all particle Euler angles were adapted to orient the MT lattice along the z-axis by matching the 2D classes at different rotations to a projection (implemented in RELION (Zivanov et al, 2018)) of a synthetic reference 13 protofilament MT (Cook et al, 2020) by cross-correlation. Cross-correlation and rotation was performed using PyTom (Hrabe et al, 2012). Next, particles were re-extracted after recentering onto refined particle coordinates and subjected to 3D refinement for initial centring of the particles, using a cryo-EM density of the native isolated *Xenopus laevis* γ-TuRC (EMDB 10491) (Liu et al, 2020) as a reference, from which density for GRIP2 domains and γ-tubulins of spoke 5 and 6 where removed using the Segger tool (Pintilie et al, 2010) implemented in UCSF Chimera (Pettersen et al, 2004), while the solvent mask only covered the γ-TuRC (Fig. EV1A). During this step, --sigma_tilt and --sigma_psi arguments were set to 0.5, to allow global orientational searches only around the MT axis, while limiting the angular sampling range around the remaining axes to local searches.

Resolution during this initial refinement step was not sufficient to enable alignment on individual spokes in correct spoke register,

as evident from low-significance density for additional spokes extending over the ends of the γ-TuRC spiral. We therefore opted to supplement 31,216 polished particles of native, isolated, non-nucleating γ-TuRC from *Xenopus laevis* (datasets 1–3 from Liu et al (2020), at 2.1 Å/px). Firstly, these particles were refined in 3D in the absence of minus end particles to orient particles according to the aforementioned reference density. Density for GRIP2 domains and γ-tubulins of spoke 5 and 6 was then subtracted from all isolated γ-TuRC particles, to serve as a control for reference bias (Fig. EV1A,C).

Capped MT minus ends were re-extracted at 2.1 Å/px to match the isolated γ-TuRC particles. Tilt and psi angles, but not translations and phi angles (i.e., rotation around the MT axis) were reset to the values obtained after 2D classification. Capped MT minus ends were refined together with supplemented isolated γ-TuRCs, keeping all other parameters as in the previous refinement step. As refinement now reached subnanometer resolution, particles could align on detailed features such as individual GCP-γ-tubulin spokes. After refinement, capped MT minus ends were split from supplemented particles and post-processed independently, reaching a resolution of 17 Å (Fig. EV1D, after pixel size correction, see below) for the γ-TuRC at the MT minus end. Notably, the GRIP2 domains and γ-tubulins of spokes 5 and 6, which were subtracted from the reference and all supplemented particles of the isolated γ-TuRC, were well-resolved in the capped minus end density (Fig. EV1C), indicating that minus end particles have been correctly aligned without bias by the reference or supplemented particles. This notion was corroborated by the observation of conformational changes and additional density segments not present in the reference or added particles.

To better resolve the MT density, capped minus ends were then subjected to an additional refinement step in the absence of supplemented particles and with a solvent mask expanded to include the first layer of α/β-tubulin dimers in the MT, allowing only local angular searches. As the MT dominated the refinement, expanding the mask was sufficient to improve the MT density. The refined density was post-processed with a mask covering only the MT, resulting in a resolution of 23 Å (Fig. EV1D, after pixel size correction, see below), yielding well-defined density for 13 protofilaments.

Using external reference data recorded on the same microscope and at the same magnification, we established that the actual pixel size at the used magnification is 2.54 Å/px and not 2.66 Å/px, as was used during data processing. Therefore, we post-processed all outcome densities taking into account the correct pixel size.

## Cryo-EM data processing for γ-TuRC-capped MTs not subjected to shortening

6996 candidate particles of γ-TuRC-capped MTs not subjected to shortening during purification (stabilised with Paclitaxel) were manually picked and combined with 9382 (Paclitaxel) and 8155 particles (DTX, including one dataset not used for the processing described above due to lower particle quality). Particles were sorted by 2D classification as described above, leaving 4626 particles of γ-TuRC-capped MTs not subjected to shortening. These particles were treated separately from the first 3D refinement step onwards. Subsequently, all processing was performed as described above.

## Cryo-EM data processing to reconstruct the MT lattice

To reconstruct the MT lattice distant from the γ-TuRC in the Paclitaxel dataset, we manually picked start-end coordinates in RELION 3.1 (He and Scheres, 2017) of straight, non-crossing and uncontaminated MT stretches for all γ-TuRC-capped MTs retained after 2D classification (see above). A total of 33,311 MT segments were extracted at full spatial resolution (2.54 Å/px) from start-end coordinates with a helical rise of 82 Å and 236 px box size. Without any sorting steps, MT segments were refined with the application of helical symmetry (13 unique asymmetrical units, −27.67 degrees initial twist and 9.46 Å rise) and post-processed, resulting in a reconstruction at 6.4 Å resolution.

## Protofilament number determination

The protofilament number of 33 high-quality capped MT minus end particles from the Paclitaxel dataset, selected from the pool of particles retained after 2D classification (see section above) was determined by supervised 3D classification in RELION 3.1 (Zivanov et al, 2018) based on the protocol outlined by Cook et al (2020), with some modifications. In brief, straight, uncontaminated and non-crossing stretches of MT directly adjacent to the γ-TuRC cap were manually picked and extracted as helical segments with a helical rise of 82 Å at 5.08 Å/px, taking into account the corrected pixel size of 2.54 Å/px. Then, the particles were subjected to a single iteration of supervised 3D classification using synthetic references of MTs with different protofilament numbers low-pass filtered to 15 Å, as provided by Cook et al (2020). Consensus protofilament number was then determined by the protofilament number assigned to the majority of segments. In the vast majority of cases (30 of 33), at least 90% of segments were assigned the same protofilament number.

The protofilament number of the same MTs was also assigned through analysis of their Moiré pattern (Chretien et al, 1996; Chretien and Wade, 1991). Visualisation of Moiré patterns was aided by orienting the MT of interest vertically, followed by successive Fourier transformation, application of a rectangular mask around the origin and inverse Fourier transformation (Atherton et al, 2018) of a cut-out of the micrographs covering only the MT of interest in ImageJ 2.0 (Rueden et al, 2017).

## Rigid-body fitting of atomic models for the γ-TuRC

For the γ-TuRC, atomic models corresponding to individual spokes of the isolated *Xenopus laevis* γ-TuRC (PDB 6TF9 (Liu et al, 2020)) were docked into the post-processed density of the γ-TuRC at the MT minus end as rigid bodies based on cross-correlation, using the Fit in Map procedure in UCSF Chimera (Pettersen et al, 2004). To create an atomic model for the lumenal bridge (without actin), models for *X. laevis* GCP3$^N$/MZT1 and GCP6$^N$/MZT1 were predicted and relaxed using AlphaFold Multimer (preprint: Evans et al, 2022; Jumper et al, 2021) v2.3 as implemented in ColabFold (Mirdita et al, 2022). The top-ranked model of each prediction was trimmed and fitted into an 8 Å molmap of the respective helices in the isolated *Xenopus laevis* γ-TuRC (PDB 6TF9 (Liu et al, 2020)), which were only built as pseudoatomic model. This actin-less lumenal bridge was then rigid-body docked into the reconstruction of the γ-TuRC at the

MT minus end as a single unit. Density for actin was insufficiently resolved and hence not modelled.

## Geometric analysis of the γ-TuRC

To generate an atomic model for the hypothetical fully closed vertebrate γ-TuRC, we first created an atomic model for 14 spokes of the budding yeast γ-TuSC ring. For this, we rigid-body docked all γ-tubulin and GCP molecules (GRIP1 and GRIP2 domain only) of the isolated *X. laevis* γ-TuRC (PDB 6TF9 (Liu et al, 2020)) into the cryo-EM density of the helical γ-TuSC assembly in the closed conformation (EMDB 2799 (Kollman et al, 2015)) on a per subunit basis. The resulting model was used to define the γ-TuRC axis for all further analyses, i.e., the axis through the centroids of the following atoms (based on Liu et al (2020)): centroid 1—γ-tubulin Thr145 of spoke 1 to 13, centroid 2—γ-tubulin Tyr152 of spoke 1 to 13, centroid 3—Ser369 of GCP2, Pro408 of GCP3, Pro166 for GCP4, Glu451 for GCP5 and Tyr445 of GCP6 for spoke 1 to 13, centroid 4—Leu216 of GCP2, Leu249 of GCP3, Met1 of GCP4, Val266 of GCP5 and Leu280 of GCP6 for spoke 1 to 13. Atomic models for the isolated γ-TuRC (PDB 6TF9 (Liu et al, 2020)) and γ-TuRC at the MT minus end (described in previous paragraph) were rigid-body docked into a molmap (generated at 10 Å resolution) of the GCP GRIP1 and GRIP2 domains as well as γ-tubulins of spokes 3 to 9 of the hypothetical fully closed γ-TuRC, where conformational changes between different states were minimal. Helical pitch increments, rotations around the helical axis and distances to helical axis were calculated using the centres of mass of γ-tubulins in each state in UCSF Chimera (Pettersen et al, 2004).

## Structure validation of the γ-TuRC model

To further validate our cryo-EM reconstruction, we systematically compared cross-correlation between cryo-EM densities obtained using various approaches and atomic models of the *open* (PDB 6TF9), *closed* (prepared as described above) and *partially closed* conformations of the γ-TuRC (see Table EV1, Appendix Tables S2–4). To enable comparison, only chains present in all models were used for fitting and cross-correlation in UCSF Chimera (Pettersen et al, 2004). To simulate particle images of the hypothetical closed γ-TuRC, the cryo-EM density of the budding yeast γ-TuSC ring (EMDB-2799 (Kollman et al, 2015)) was segmented to 14 spokes, leaving out the GRIP2 domains and γ-tubulins of spoke 5 and 6, and projected with white noise using relion_project to produce 31,216 particle images with the same orientations and CTF parameters as the original supplemented particles of the isolated open γ-TuRC. As an additional validation step, the original joint refinement supplemented with particle images of native open γ-TuRC was followed by a joint refinement step with local angular sampling and a solvent mask covering only spokes 1 to 8. Finally, we performed a step of global refinement without any particle supplementation following the refinement with supplemented isolated γ-TuRC (Fig. EV2A–C).

## Rigid-body fitting of atomic models for the γ-TuRC-associated MT

The density of the capped MT was sufficient for resolving individual protofilaments, but the high rotational symmetry along

the protofilament axis at moderate resolutions precluded mere rigid-body fitting of individual protofilaments into the density. We first generated an atomic model of a protofilament based on PDB 6WVM (Debs et al, 2020) by matchmaking two copies of this model onto the protofilament model of a Paclitaxel-stabilised MT (PDB 6EW0) (Manka and Moores, 2018). This model was rigid-body docked into each protofilament of the reconstruction of the minus end of the γ-TuRC-capped MT. Each protofilament was then rotated around its own axis iteratively to minimise the distance at the inter-protofilament contact, defined as residues β-tubulin Tyr283 and Ala56. Lastly, the position of the protofilaments along the MT axis was estimated based on cross-correlation with the cryo-EM density as a function of displacement along its own axis. To aid this, particles with offsets and orientations derived from MT-dominated refinement were re-extracted in larger boxes (256 px), followed by reconstruction and post-processing. The resulting cross-correlation pattern contains peaks representing correlation of the protofilament model with the density in different tubulin registers. The correct position was determined by simultaneous fitting of multiple Gaussian curves based on an implementation by Emily Ripka (Github: https://github.com/emilyripka/BlogRepo/blob/master/181119_PeakFitting.ipynb). Raw cross-correlation patterns and fitted curves are available as source data.

### Geometric analysis of the emanating microtubule

A cryo-EM density corresponding to three MT protofilaments of a 13 protofilament MT (Manka and Moores, 2018) (EMDB 3965) was expanded step-wise until a 13 protofilament MT density was formed. The associated atomic model (PDB 6EW0) was then fit in to create a 13 protofilament MT atomic model, which was fit into a molmap at 24 Å of protofilaments at spoke 3 to 7 of the MT at the capped minus end. The MT axis was then defined as the axis through centroids of Gln15 for each of the tubulin layers in the reference 13 protofilament MT. Lateral offsets (i.e., perpendicular to the MT axis) between γ- and α-tubulin were determined based on the centre of mass of γ-tubulin and the closest α-tubulin in the associated protofilament after rigid-body docking the previously described model of the partially closed γ-TuRC into the cryo-EM density map into which the MT was modelled; for the determination of other MT parameters, the centre of mass of the bottom α-tubulin was used.

## Data availability

The datasets and computer code produced in this study are available in the following databases: (1) Atomic coordinates and the associated cryo-EM densities have been deposited to the Protein Data Bank and the Electron Microscopy Data Bank: PDB-9EOJ, EMD-19861, PDB-9EOK, EMD-19862. (2) Previously published structural data used in this paper: PDB-6EW0, PDB-6TF9, PDB-6WVM, PDB-7QJC, EMD-10491, EMD-2799, EMD-3965. (3) The used GFP-CAMSAP2 construct is available from Addgene [Plasmid #59037].

## Peer review information

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

## Acknowledgements

We thank Mandy Rettel (Proteomics Core Facility, EMBL Heidelberg) for performing mass spectrometry analysis. Moreover, we thank Sebastian Eustermann (EMBL Heidelberg) and Carsten Janke (Institut Curie, Orsay) for insightful discussions. We are grateful to Sebastian Filbeck, Sophie Kopetschke (both ZMBH, Heidelberg University) and Jan-Philipp Kreysing (MPI of Biophysics, Frankfurt am Main) for assistance with data processing. Furthermore, we thank Analena Lefèvre (Institut für Genetik, Universität Bonn) for help in preparing egg extracts. We acknowledge access to the infrastructure of the Cryo-EM Network at Heidelberg University (HDcryoNET) and support by Götz Hofhaus (BioQuant, Heidelberg University) and Dirk Flemming (BZH, Heidelberg University). Moreover, we acknowledge the EM Core Facility at Heidelberg University for access to their infrastructure and thank Dr. Charlotta Funaya for helpful discussions. We also acknowledge the services SDS@hd and bwHPC supported by the Ministry of Science, Research and the Arts Baden-Württemberg, as well as the German Research Foundation (INST 35/1314-1 FUGG, INST 35/1503-1 FUGG and INST 35/1134-1 FUGG). This work is supported by grants of the Deutsche Forschungsgemeinschaft (DFG) to ES (DFG Schi 295/4-4) and to SP (DFG PF 963/1-4). SP also

acknowledges funding by the Aventis Foundation and the Chica and Heinz Schaller Foundation.

## Author contributions

**Bram JA Vermeulen**: Conceptualization; Data curation; Software; Formal analysis; Validation; Investigation; Visualization; Methodology; Writing—original draft; Project administration; Writing—review and editing. **Anna Böhler**: Conceptualization; Data curation; Formal analysis; Validation; Investigation; Visualization; Methodology; Writing—review and editing. **Qi Gao**: Data curation; Formal analysis; Validation; Investigation; Visualization; Methodology. **Annett Neuner**: Data curation; Formal analysis; Investigation; Visualization; Methodology; Writing—review and editing. **Erik Župa**: Investigation; Writing—review and editing. **Zhenzhen Chu**: Resources. **Martin Würtz**: Data curation; Formal analysis; Validation; Investigation; Visualization; Writing—review and editing. **Ursula Jäkle**: Investigation. **Oliver J Gruss**: Conceptualization; Resources; Methodology; Writing—review and editing. **Stefan Pfeffer**: Conceptualization; Data curation; Supervision; Funding acquisition; Validation; Methodology; Writing—original draft; Project administration; Writing—review and editing. **Elmar Schiebel**: Conceptualization; Data curation; Supervision; Funding acquisition; Validation; Methodology; Project administration; Writing—review and editing.

## Funding

## Disclosure and competing interests statement

The authors declare no competing interests.

# Expanded View Figures

**Figure EV1.   Cryo-EM processing workflow.**

(**A**) Detailed image processing scheme. Colours of text and reconstructions indicate the particles used for reconstruction; capped MT minus ends in green, isolated γ-TuRCs in red, a mixture of both in khaki and reference densities in yellow. Respective Fourier Shell Correlation (FSC) curves for resolution values marked by orange and green line are shown in panel (**D**). (**B**) Sample cryo-EM micrograph of γ-TuRC-capped MTs from *Xenopus laevis*, with particles retained for final reconstruction indicated. Scale bar represents 50 nm. (**C**) Comparison of the reconstruction of the γ-TuRC at the MT minus end (orange, left) and the reconstruction of isolated γ-TuRC particles (grey, right) from the same joint refinement run, showing the former reconstruction is not biased by supplementation of isolated γ-TuRC during refinement. Outlined box shows the presence of density for the GRIP2 domain and γ-tubulin of spokes 5 and 6 (indicated by a box) in the γ-TuRC at the MT minus end. This density was subtracted from the supplemented particle images of isolated γ-TuRC and removed from the isolated γ-TuRC reconstruction that served as a reference for refinement. To enable a fair comparison, both reconstructions are low-pass filtered to 17 Å, the resolution of the reconstruction for the γ-TuRC at the MT minus end. Spoke numbering and colouring scheme are indicated. Density for spokes in the background was hidden (clipped) for visual clarity in both panels. (**D**) FSC curve for the γ-TuRC at the MT minus end from γ-TuRC-focused refinement, after removal of supplemented particles (orange) and FSC curve for the MT-dominated local refinement after γ-TuRC-focused refinement (green). Both curves were calculated taking into account a corrected, unbinned pixel size of 2.54 Å/px (see Methods). Threshold at FSC = 0.143 is indicated by a dashed line; the resolution at which the FSC curve first passes the threshold is specified. Source data are available online for this figure.

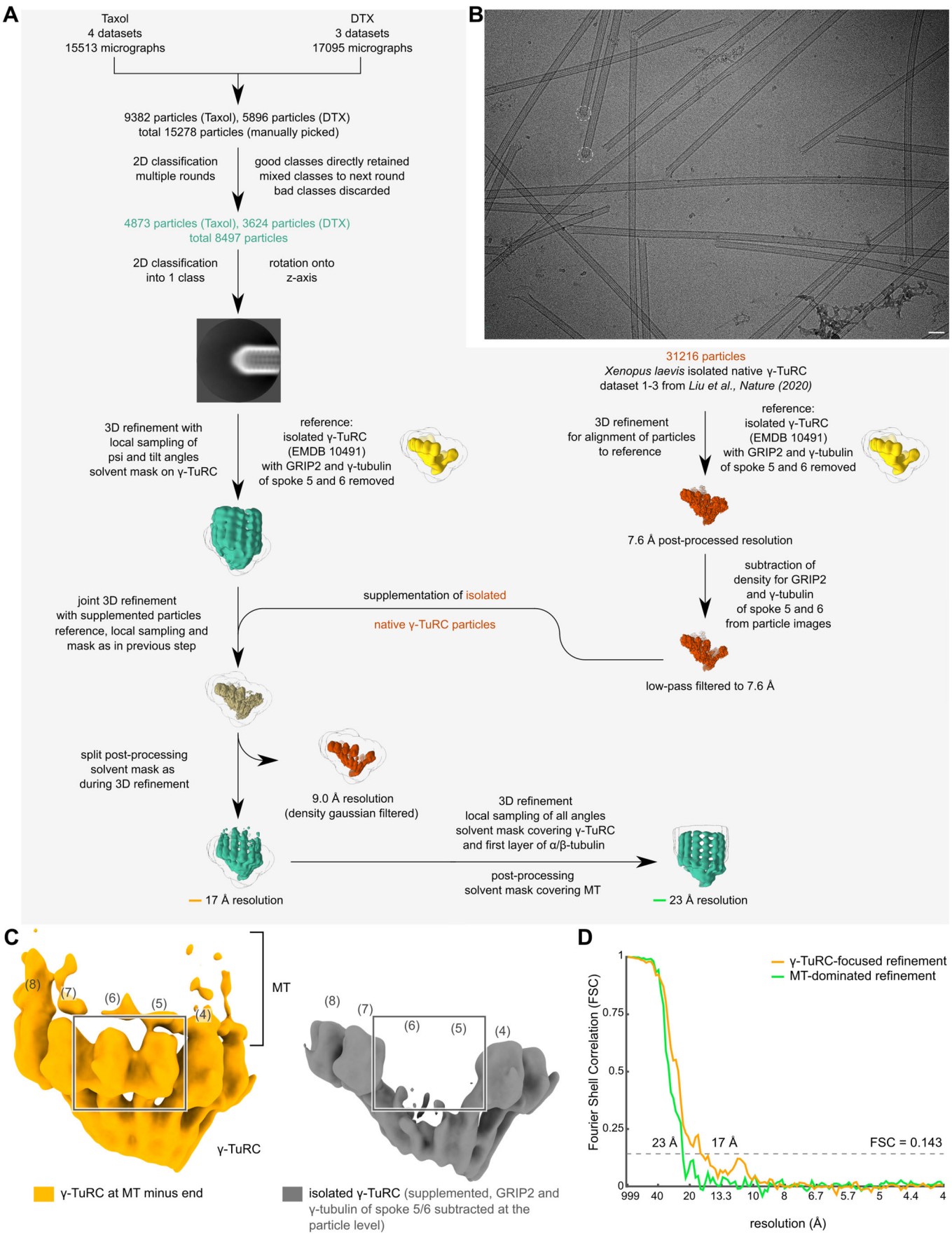

**C**

γ-TuRC at MT minus end

isolated γ-TuRC (supplemented, GRIP2 and γ-tubulin of spoke 5/6 subtracted at the particle level)

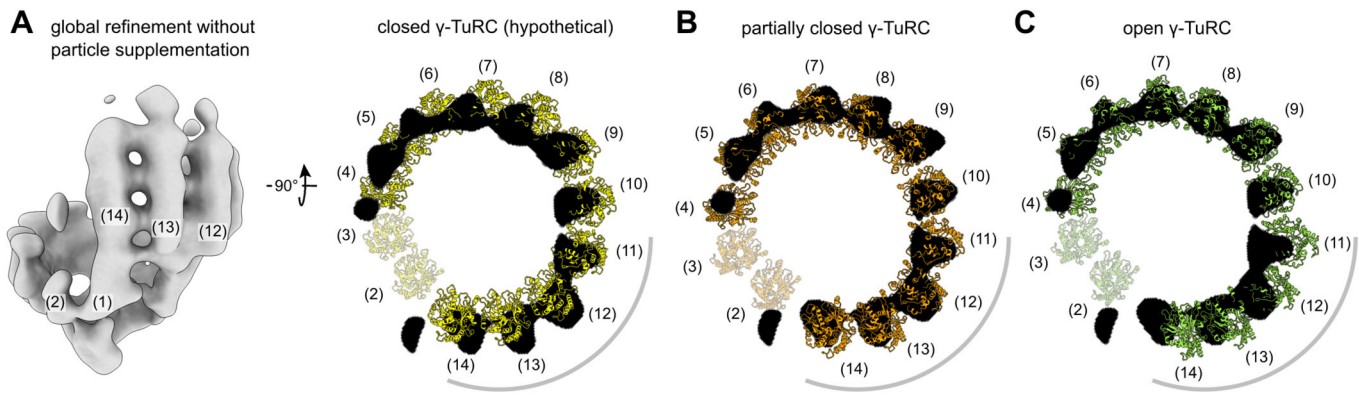

slice through reconstruction without particle supplementation

**Figure EV2.   Refinement and reconstruction of the MT-capping γ-TuRC without particle supplementation confirms a partially closed conformation.**

(A–C) Refinement of only MT-capping-γ-TuRC particles at global angular sampling, without any particle supplementation, following refinement with particle supplementation. Atomic models for the hypothetical closed γ-TuRC (Kollman et al, 2015) (A), the partially closed γ-TuRC (this study, B) and the open γ-TuRC (PDB 6TF9 (Liu et al, 2020), C) were then rigid-body docked based on the GRIP1/2 domains and γ-tubulin molecules of spokes 3 to 9. The partially closed γ-TuRC (B) fits the reconstruction well, in spite of the absence of supplemented particles during the refinement, as opposed to the closed (A) and open (C) γ-TuRC, which clearly deviate from the reconstruction, especially at spokes 11 to 14 (highlighted by a grey line). This shows that the particle supplementation does not induce bias towards the open conformation. For clarity, only a slice through the reconstruction is shown, superposed with γ-tubulin molecules 2–14 of the fit model. The slice was chosen to highlight the fit in the upper half of the spokes. As the slice does not cover spokes 2 and 3, they are shown transparent. Spoke numbering is indicated.

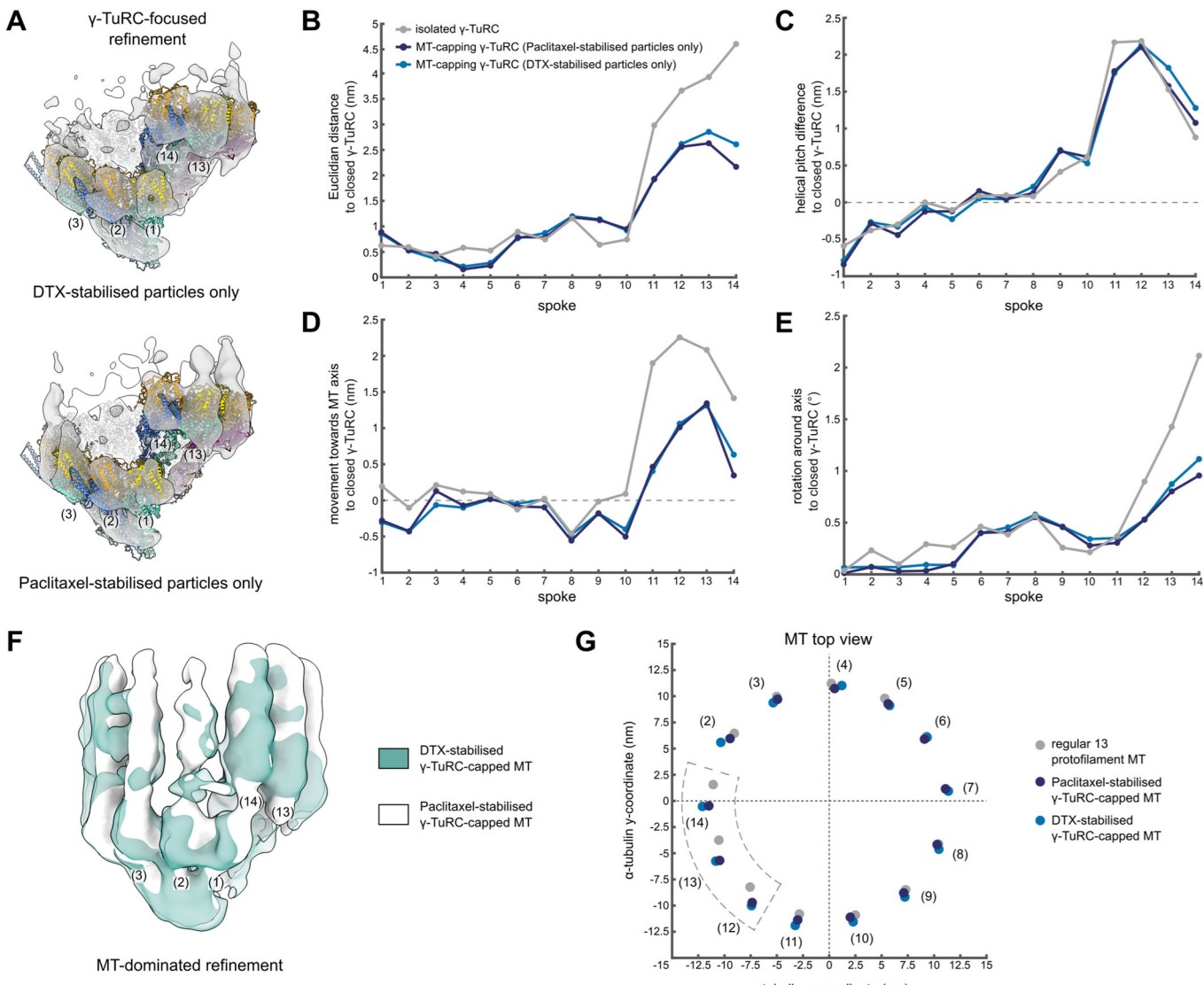

**Figure EV3. The conformation of the γ-TuRC and attached MT is not affected by the identity of the MT stabilisation agent.**

(A) Reconstruction obtained from γ-TuRC-focused refinement of MT-capping γ-TuRCs with only DTX-stabilised particles (top) or only Paclitaxel-stabilised particles (bottom). A model for the γ-TuRC was generated for each reconstruction separately by spoke-wise rigid body docking. A high confidence threshold is used to highlight the fit at the level of γ-tubulin. Spoke numbering is indicated. Colouring as in Fig. 2A. (B–E) Euclidian translation distance (B), downward change in helical pitch (C), translation towards the MT axis (D) and rotation around the MT axis (E) required to convert the models to the hypothetical, fully closed γ-TuRC for each spoke. Parameters measured from the centre of mass of γ-tubulin. (F) Reconstructions of MT-dominated refinements of γ-TuRC-capped MTs with only DTX-stabilised (green) or only Paclitaxel-stabilised particles (white). Density for spokes in the background was hidden for visual clarity. Colouring scheme is indicated. (G) Coordinates of the centre of mass for the first α-tubulin in each protofilament in the plane orthogonal to the MT axis for regular 13 protofilament lattice (grey), Paclitaxel-stabilised γ-TuRC-capped MT (dark blue) and DTX-stabilised γ-TuRC-capped MT (light blue). Protofilaments are numbered by the respective spokes in the γ-TuRC. The MT axis is placed on the origin, (0,0). Colouring scheme is indicated. Source data are available online for this figure.

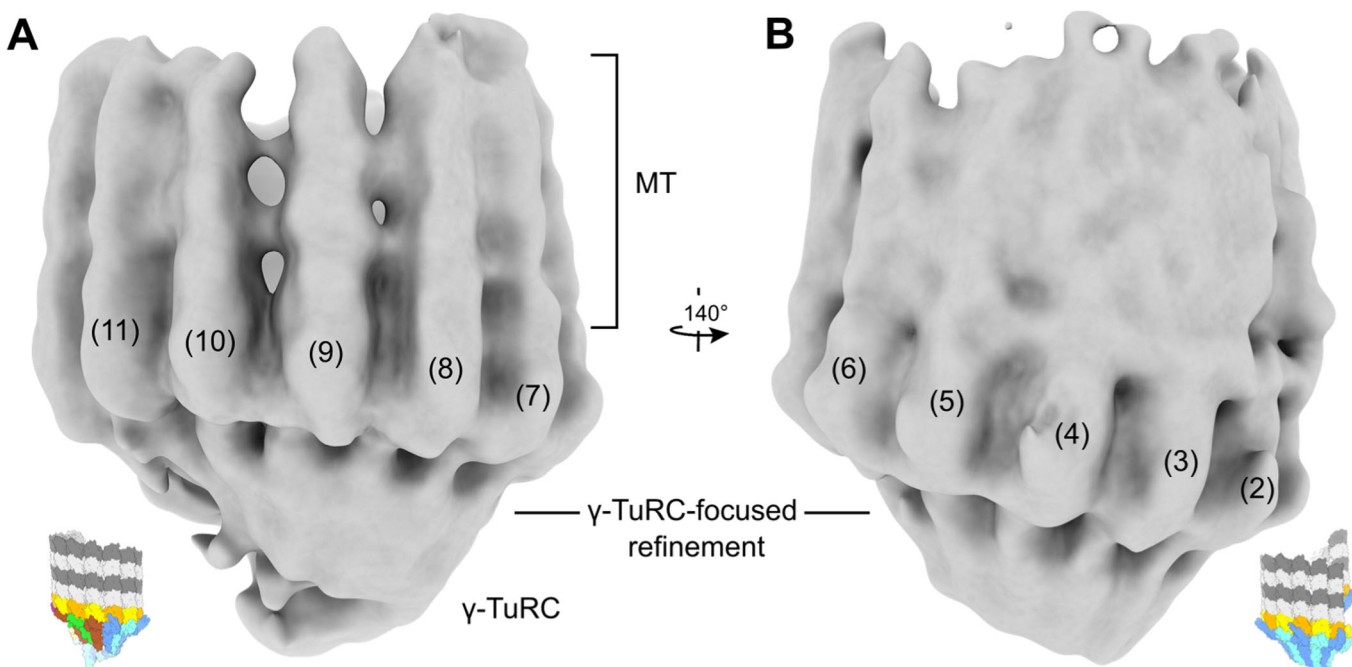

**Figure EV4. Not all MT protofilaments are equally rigid with respect to the γ-TuRC.**

(A, B) Refinement focused on the γ-TuRC, in which all γ-TuRC spokes are well-defined, indicating that the γ-TuRC was well-aligned all around. In contrast, the MT density shows local variations in definition: protofilaments associated with spokes 7–11 of the γ-TuRC (A) are considerably better defined than those associated with spokes 2–5 (B). This indicates increased conformational plasticity of the MT relative to the γ-TuRC at the symmetric side of the γ-TuRC. Reconstructions are local resolution-filtered, shown at low threshold to emphasise MT density. Spoke numbers are indicated. Inset schematics show the orientation of the γ-TuRC.

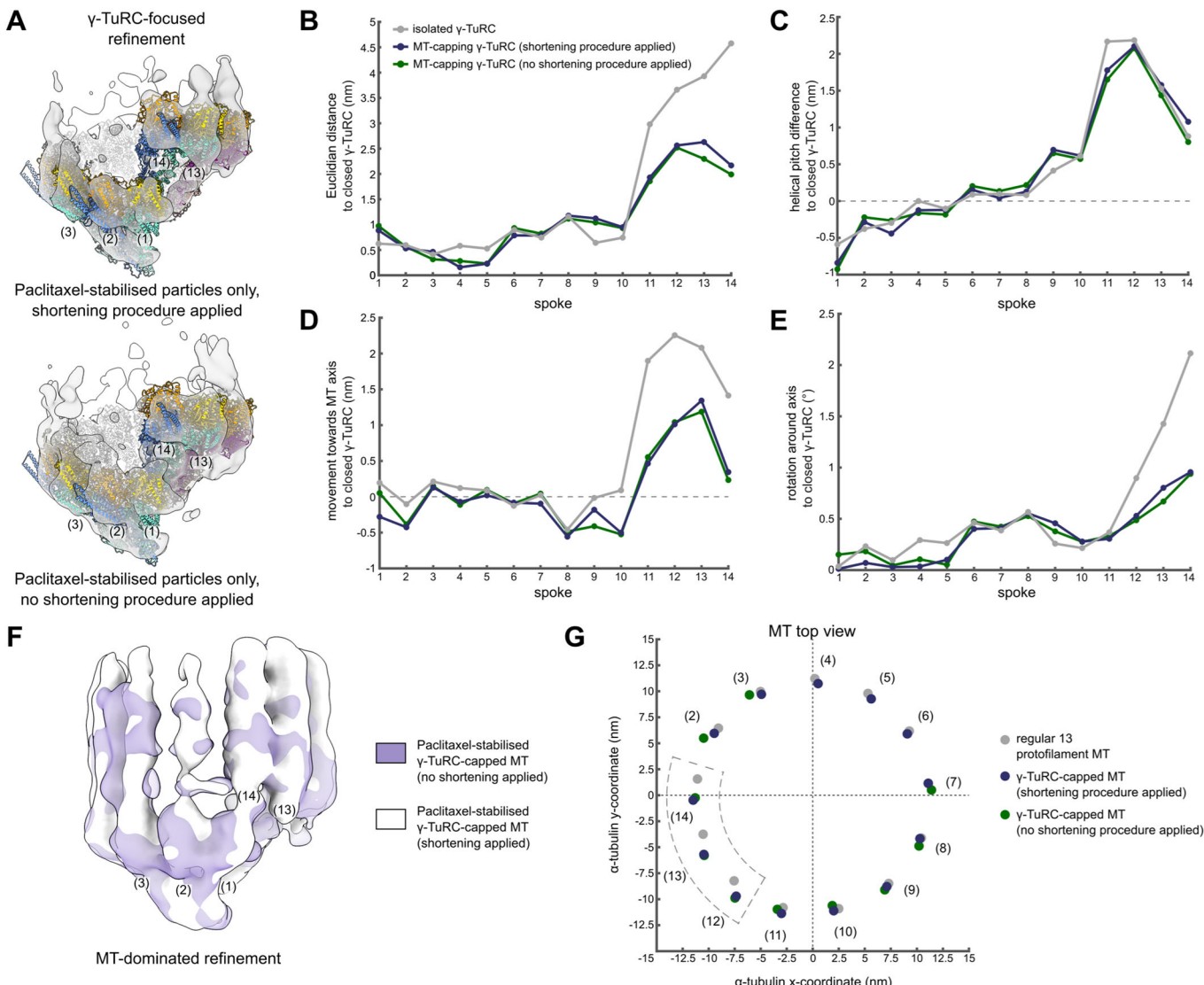

**Figure EV5.  The conformation of the γ-TuRC and attached MT is not affected by the shortening procedure applied during purification.**

(A) Reconstruction obtained from γ-TuRC-focused refinement of γ-TuRCs capping Paclitaxel-stabilised microtubules which were (top) or were not (bottom) subjected to the shortening procedure during purification. A model for the γ-TuRC was generated for each reconstruction separately by spoke-wise rigid body docking. A high confidence threshold is used to highlight the fit at the level of γ-tubulin. Spoke numbering is indicated. Colouring as in Fig. 2A. (B–E) Euclidian translation distance (B), downward change in helical pitch (C), translation towards the MT axis (D) and rotation around the MT axis (E) required to convert the models to the hypothetical, fully closed γ-TuRC for each spoke. Parameters measured from the centre of mass of γ-tubulin. (F) Reconstructions obtained from MT-dominated refinements of γ-TuRC-capped Paclitaxel-stabilised MTs which were (white) or were not (purple) subjected to the shortening procedure during purification. Density for spokes in the background was hidden for visual clarity. Colouring scheme is indicated. (G) Coordinates of the centre of mass for the first α-tubulin in each protofilament in the plane orthogonal to the MT axis for the regular 13 protofilament lattice (grey), the Paclitaxel-stabilised γ-TuRC-capped MT where shortening was (dark blue) or was not (green) applied during purification. Individual protofilaments could not be fitted at spokes 4 to 6 for the condition without shortening due to lower density quality. Protofilaments are numbered by the respective spokes in the γ-TuRC. The MT axis is placed on the origin, (0,0). Colouring scheme is indicated. Source data are available online for this figure.

