## [Peer Review File · The EMBO Journal]

γ -TuRC asymmetry induces local protofilament mismatch at the RanGTP-stimulated microtubule minus end

Elmar Schiebel, Bram Vermeulen, Anna Böhler, Qi Gao, Annett Neuner, Erik Župa, Zhenzhen Chu, Martin Würtz, Ursula Jäkle, Oliver Gruss, and Stefan Pfeffer

Corresponding author(s): Elmar Schiebel (e.schiebel@zmbh.uni-heidelberg.de) , Stefan Pfeffer (s.pfeffer@zmbh.uni-heidelberg.de)

Review Timeline:

Submission Date:	3rd Nov 23
Editorial Decision:	14th Dec 23
Revision Received:	16th Feb 24
Editorial Decision:	11th Mar 24
Revision Received:	15th Mar 24
Accepted:	18th Mar 24

Editor: Hartmut Vodermaier

Transaction Report:

Referee reports for EMBOJ-2023-116073

Dear Elmar and Stefan,

Thank you again for submitting your manuscript on native gamma-TuRC structure to The EMBO Journal. We have now received a complete set of reviews from three expert referees, which I am copying below for your information. As you will see, the referees all express interest in the work, but raise a variety of concerns that may affect the conclusiveness of the study. Before deciding how to best proceed further with the study, I would like to give you an opportunity to consider these reports, and to get back to me with a tentative point-by-point letter explaining how you would answer the various points. The most important issue would obviously be validity of the challenging aspects connected to this structural determination. Please note that we would of course not require you to reconcile your findings with other, parallel studies that were pre-printed after submission of your own manuscript; but it will still be interesting to discuss and possibly rationalize apparent differences, as asked especially by referee 2.

It would be great if you could get back to me with a tentative response early next week, on the basis of which I would be happy to schedule an informal Zoom call to talk through the main points, before taking a final decision on the manuscript.

Best regards,

Hartmut

Hartmut Vodermaier, PhD

Senior Editor, The EMBO Journal

h.vodermaier@embojournal.org

Referee #1 (Report for Author)

In this paper, Vermeulen et al. present a 17 Angstrom resolution structure of the *Xenopus* γ -TuRC associated with the minus end of a nucleated microtubule. This goes beyond what has been achieved previously in higher Eukaryotes, where structures of unattached γ -TuRCs have been elucidated. These unattached γ -TuRCs were shown to have a semi-open conformation where γ -tubulin 1 and 14 did not overlap and where the helical pitch did not match that of a microtubule. Thus, an important question has remained as to whether γ -TuRCs change conformation during or before nucleation. To address this, the authors stimulated the nucleation process within *Xenopus* egg extracts by adding constitutively active RanGTP and then affinity-purified γ -TuRCs bound to microtubules. The authors surprisingly find that the microtubule-attached γ -TuRCs do not fully

"close" resulting in mismatches between γ -tubulin and microtubule protofilaments. The largest mismatches occur at positions 2-7. Moreover, the protofilament from spoke 2 has clear gaps between the protofilaments coming from spokes 3 and 14, except for a small connection to protofilament 14 that may be via the GCP3/Mzt1 module previously observed in a similar position. The authors suggest that the asymmetry in protofilaments may provide selective binding for CAMSAP2, potentially explaining how it may recognise minus ends capped by γ -TuRCs, although this was not formally tested. The authors also suggest that the imperfect match allows tubulin concentration to have a greater impact on the nucleation process. The authors explained that there were a limited number of particles to analyse due to the native source of the γ -TuRCs, and so they used a "tailored multistep approach for data processing". This reviewer is not an expert in cryo-EM single particle analysis and so cannot properly comment on the validity of this approach, although from a non-expert view it seems reasonable and necessary. Overall, this is an excellent paper and makes a conceptual advance to the field.

Major comments

- 1) The mass spec data shows that Augmin/HAUS is associated with the purified γ -TuRC-microtubule complexes. It was recently shown that Ran-GTP stimulates Augmin/HAUS-mediated microtubule nucleation (Kraus et al., 2023, JBC). Thus, it seems that the pathway of nucleation the authors are studying is Augmin-mediated. This should be explicitly stated in the results and perhaps discussed in terms of how this influences γ -TuRC conformation and activity, noting that a couple of papers have already concluded that Augmin binding increases γ -TuRC activity.
- 2) Similar to point 1, the mass spec data also shows the absence of CDK5RAP2 (CM1) domain. This is an important point, and, while touched upon in the discussion, I think the authors need to make this clear to the reader in the results section.
- 3) When the authors analyse the seam of the microtubule, they show the microtubule lattice >100nm from the γ -TuRC, stating that the microtubule lattice is regular. But do they know that the seam has formed correctly? They don't appear to look at the seam directly. Moreover, I don't see how this shows that their "results suggest that the position of the MT seam can be pre-determined by the orientation of the γ -TuRC template, even before MT nucleation starts" as they don't directly show which protofilaments the seam forms between. They just show that it would take less movement to form between protofilaments 2 and 14.

Minor comments

- 1) In the introduction, the authors say: "In vivo, however, MT nucleation is templated by γ -tubulin complexes (γ -TuCs), yielding MTs with a defined number of 13 protofilaments (Tilney et al, 1973)." This is an overstatement. Firstly, many cells have been shown to have microtubules with different protofilament numbers (see Review by Gary Brouhard). Secondly, the way the statement is made

suggests that we already know that γ -TuRC-mediated nucleation produces 13-protofilament microtubules. But at this stage in the paper, we don't know whether the γ -TuRC will change conformation during nucleation and so could in theory produce a 14-protofilament microtubule.

2) In the introduction, the authors mention GCP2/Mzt2 modules and say that they have only been observed when bound to the CM1 domain. Firstly, is it known that this module binds the CM1 domain, or are they just close in space? Secondly, the authors should mention the work from the Luders lab that provides good evidence for Mzt2-GCP2-NTE modules decorating the outside of the γ -TuRC at GCP2/3 interfaces (Zimmerman et al).

Referee #2 (Report for Author)

Microtubule (MT) dynamics are centrally involved in numerous cell activities, and the regulated addition and loss of tubulin subunits at both MT plus and minus ends is thus a critical aspect of cell physiology. A molecular level understanding of the architecture of MT ends, and of regulators of end dynamics, therefore provides important mechanistic insight into cellular processes. γ -Tubulin Ring Complexes (g-TuRCs) both nucleate MTs, cap and stabilize MT minus ends, and their distribution within cells defines the organization of the MT array.

New information has recently been revealed about the structures of inactive g-TuRCs from a number of different eukaryotes using cryo-EM. As well as enabling direct and detailed visualization of these multi-protein complexes, this important set of studies provided suggestive insights as to how MT nucleation might occur on activation. However, precise experimental data about the structure of MT minus ends capped by g-TuRCs has been missing.

The study by Vermeulen et al seeks to address this knowledge gap. The authors mainly use tubulin-rich *Xenopus* egg extract to purify g-TuRC-nucleated MTs, which are drug-stabilized by either taxol or docetaxel. The purification protocol involves both "mild mechanical shearing" of longer MTs in the sample before affinity purification of the g-TuRCs (and attached shorter MTs) and subsequent cryo-EM grid preparation. It is therefore likely that these stabilizing drugs were important during sample preparation to maintain the integrity of the otherwise dynamic MTs. Because of challenges with the sample, the authors present an analysis of modestly-sized datasets of taxol- or docetaxel-stabilized MTs, and ultimately (and apparently legitimately) merge these datasets to achieve their final reconstruction of the g-TuRC with attached MT.

Albeit visualized at modest resolution (~20 Å), the authors' structure very surprisingly, does not display an exact match between the g-TuRC and the pseudo-helical structure of the MT that emerges from it. Rather, a patch of structural discontinuities is observed centred around g-tubulins

1 and 14, which do not perfectly align, and which renders the MT minus end slightly non-cylindrical, or as the authors describe it "profoundly asymmetric".

2 key questions arise from this finding which the authors need to address:

- 1) Does the reported asymmetric structure reflect the active *Xenopus* g-TuRCs in their samples?
- 2) What does the reported structure tell us about MT nucleation and capping mechanisms?

Question 1

The image processing scheme used by the authors involves inclusion of images of inactive g-TuRCs to drive 3D alignment. This is a counter-intuitive strategy given the aim of determining the structure of active g-TuRC. According to the authors' account, it seems that without the probably more overtly asymmetric inactive g-TuRC data, the small dataset of more symmetric active g-TuRCs would not align convergently. The inactive g-TuRC data is not included in the final reconstruction, and some evidence is provided (EV Fig 4C) that at least some crude features of the inactive g-TuRC data are not carried through into the final active g-TuRC + MT reconstruction. However, concerns remain that the asymmetry of the inactive g-TuRC data has been imprinted on the final reconstruction such that reported asymmetry of the MT end arises from it. There is a hint this might be the case from the data EV Fig 8B, in which more symmetric regions of the capped minus end are said to be more plastic, but this could equally be because they are less well aligned. EV Fig 5 is also presented as evidence of a lack of bias in the 3D reconstruction, but if I have understood the data, this figure simply confirms that the presented reconstruction does not match either models of a hypothetical symmetric activated g-TuRC or of inactive g-TuRC.

Question 2

Given the unexpected mismatch between the g-TuRC template and the nucleated MT, the observed structure does not lead straightforwardly to mechanistic clarity concerning precise nucleation of 13 protofilament MT and capping activity. The authors propose that additional regulatory steps may exist, and don't exclude that these could be additional steps that induce full symmetrization of the g-TuRC. However, it also remains a formal possibility that the mechanical shearing step in the purification protocol could also contribute to the slightly distorted MT minus end.

Furthermore:

i) Both active yeast (<https://doi.org/10.21203/rs.3.rs-3481382/v1>) and human (<https://doi.org/10.1101/2023.11.20.567916>) g-TuRCs have been recently described at relatively higher resolution as forming symmetric structures at the minus ends of MTs. The major discrepancy between these studies and that of Vermeulen et al needs to be accounted for.

ii) The experiments relating to CAMSAP binding relate in part to recombinant human γ -TuRC about which no 3D structural information is provided. The authors can only infer that preferential CAMSAP binding is mediated via the asymmetric feature in their reconstruction. In addition, this model of CAMSAP recognition is incompletely reconciled with more detailed existing models of CAMSAP minus end binding e.g. PMID 28991265

Overall, results of this study are well presented and clearly described but there are number of critical aspects of study design and interpretation that urgently require clarification.

Referee #3 (Report for Author)

In recent years, several research groups, including those contributing to this manuscript, have elucidated the structure of the microtubule nucleator γ -tubulin ring complex (γ -TuRC) using cryo-electron microscopy (cryo-EM). The asymmetric nature of the γ -TuRC structure has prompted investigations into the regulation of this asymmetry in guiding the formation of symmetric microtubules. In this study, Vermeulen and colleagues address this question by reconstituting native γ -TuRC-capped microtubules from *Xenopus laevis* egg extract for cryo-EM analysis. The manuscript introduces a protocol for utilizing endogenous protein components of *Xenopus* egg extract to reconstitute γ -TuRC-templated microtubule nucleation through the RanGTP pathway.

Subsequent cryo-EM analysis of the γ -TuRC-capped microtubules at 17 Å reveals a partially closed ring geometry that misaligns with the emanating microtubule, deviating from the symmetry of a typical 13-protofilament microtubule. Furthermore, the γ -TuRC bound to the microtubule minus end adopts a destabilized luminal bridge, as indicated by the obscure density of the β -actin molecule. To elucidate the significance of this conformation, the authors demonstrate the interaction between CAMSAP2 and the minus end of the γ -TuRC-capped microtubule.

Increasing our comprehension of the structural compatibility of γ -TuRC as a template for microtubule nucleation and its significance in microtubule minus-end modulation, this manuscript offers valuable insights. However, before publication, we suggest addressing the following points:

1. Purification of γ -TuRC-Capped Microtubule Minus Ends: The authors employed mechanical force (extensive pipetting as described in the method section) to fragment long γ -TuRC-capped microtubules during the purification process. It is crucial to consider the potential impact of this shearing force on the structure of γ -TuRC and its capped microtubules. While the authors have demonstrated the structural integrity of the microtubule lattice away from the minus end, the preservative effect may be attributed to microtubule-specific stabilizing reagents (Paclitaxel or docetaxel), leaving the γ -TuRC-capped minus end susceptible to mechanical force.

2. RanGTP-induced microtubule formation is associated with microtubule branching, as indicated by the presence of NEDD1 and the augmin complex in the isolated γ -TuRC-capped microtubules (Expanded view table 1). This raises the possibility that the partially closed γ -TuRC conformation represents the structure unique for NEDD1/augmin-bound γ -TuRC and plays a role in microtubule branching. A more insightful discussion on these findings would enhance the manuscript.

3. While the resolution of the final density map allows rigid body fitting of the *Xenopus* γ -TuRC model, Figure 2E and Expanded view Figure 7F do not clearly illustrate the fitting of MZT1/GCP6 N terminus and MZT1/GCP3 N terminus models into the density map. This ambiguity challenges the validation of the conclusion that MZT1 was well-resolved in this study.

4. When reconstituting human γ -TuRC-capped microtubules *in vitro* for the CAMSAP2 localization assay (Figure 4A), caution is advised in assuming that the recombinant human γ -TuRC adopts a similar partially closed geometry as observed in *Xenopus* γ -TuRC-capped microtubules. Additionally, considering CAMSAP2's known binding and stabilization of free microtubule minus-ends, it may be premature to propose that CAMSAP2 recognizes the asymmetric conformation of γ -TuRC-capped microtubule minus-ends.

Minors:

1. Figure 4D presents negative staining EM micrographs illustrating microtubule minus ends capped by γ -TuRC. Unfortunately, the observation of capped minus ends is hindered by protein aggregation, making it challenging for readers. The authors should consider replacing these micrographs with higher-quality images to enhance visual interpretation.

2. In Expanded view Figure 10A, it would be helpful to include a micrograph of the entire negative staining EM grid alongside the 2D averaged classes. This will offer readers a broader perspective on the experimental setup and facilitate a more comprehensive interpretation of the results.

11.12.2023

EMBOJ-2023-116073

Point-by-Point Reply to Reviewers

We thank the reviewers for the thoughtful and supportive comments, considering this “an excellent paper” that offers “valuable insights” and “makes a conceptual advance to the field”. Below we address the specific points raised by the reviewers and elaborate on the corresponding changes in the manuscript.

Referee #1 (Report for Author)

In this paper, Vermeulen et al. present a 17 Angstrom resolution structure of the *Xenopus* γ -TuRC associated with the minus end of a nucleated microtubule. This goes beyond what has been achieved previously in higher Eukaryotes, where structures of unattached γ -TuRCs have been elucidated. These unattached γ -TuRCs were shown to have a semi-open conformation where γ -tubulin 1 and 14 did not overlap and where the helical pitch did not match that of a microtubule. Thus, an important question has remained as to whether γ -TuRCs change conformation during or before nucleation. To address this, the authors stimulated the nucleation process within *Xenopus* egg extracts by adding constitutively active RanGTP and then affinity-purified γ -TuRCs bound to microtubules. The authors surprisingly find that the microtubule-attached γ -TuRCs do not fully "close" resulting in mismatches between γ -tubulin and microtubule protofilaments. The largest mismatches occur at positions 2-7. Moreover, the protofilament from spoke 2 has clear gaps between the protofilaments coming from spokes 3 and 14, except for a small connection to protofilament 14 that may be via the GCP3/Mzt1 module previously observed in a similar position. The authors suggest that the asymmetry in protofilaments may provide selective binding for CAMSAP2, potentially explaining how it may recognise minus ends capped by γ -TuRCs, although this was not formally tested. The authors also suggest that the imperfect match allows tubulin concentration to have a greater impact on the nucleation process. The authors explained that there were a limited number of particles to analyse due to the native source of the γ -TuRCs, and so they used a "tailored multistep approach for data processing". This reviewer is not an expert in cryo-EM single particle analysis and so cannot properly comment on the validity of this approach, although from a non-expert view it seems reasonable and necessary. Overall, this is an excellent paper and makes a conceptual advance to the field.

We thank the Reviewer for this very positive evaluation of our manuscript.

Major comments

1) The mass spec data shows that Augmin/HAUS is associated with the purified γ -TuRC-microtubule complexes. It was recently shown that Ran-GTP stimulates Augmin/HAUS-mediated microtubule nucleation (Kraus et al., 2023, JBC). Thus, it seems that the pathway of nucleation the authors are studying is Augmin-mediated. This should be explicitly stated in the results and perhaps discussed in terms of how this influences γ -TuRC conformation and

activity, noting that a couple of papers have already concluded that Augmin binding increases γ -TuRC activity.

We agree with the reviewer that we may have captured the structure of a γ -TuRC-capped microtubule minus end specific for a Ran-GTP/Augmin-dependent microtubule branching pathway during spindle formation. In this pathway, thousands of microtubules are formed within a short period of time, so a mismatch between γ -TuRC and nucleated microtubules may generate a defined breaking point that promotes fast turnover of γ -TuRCs.

Importantly, we cannot exclude that γ -TuRC-capped microtubule minus ends nucleated in alternative pathways (e.g. from centrosomes) may structurally differ to accommodate differing requirements in γ -TuRC turnover and minus end stabilization. For instance, the γ -TuRC may fully close in scenarios where stable and permanent binding to the newly nucleated microtubule minus end is required.

We will discuss these aspects in more detail in the revised version of the manuscript.

2) Similar to point 1, the mass spec data also shows the absence of CDK5RAP2 (CM1) domain. This is an important point, and, while touched upon in the discussion, I think the authors need to make this clear to the reader in the results section.

The absence of CDK5RAP2 in the mass spec data goes hand in hand with the possibility of having captured the structure of a γ -TuRC-capped microtubule minus end nucleated in a native microtubule branching pathway during spindle formation, in which CDK5RAP2 is not involved. We will emphasize this aspect in the results section of the revised manuscript and extend the discussion.

3) When the authors analyse the seam of the microtubule, they show the microtubule lattice >100nm from the γ -TuRC, stating that the microtubule lattice is regular. But do they know that the seam has formed correctly? They don't appear to look at the seam directly.

A properly formed seam is a geometrical prerequisite for the formation of a regular microtubule lattice. Although we cannot directly identify the seam at the resolution achieved in our reconstruction of the microtubule lattice > 100 nm away from the minus end, we know that it must have formed.

Moreover, I don't see how this shows that their "results suggest that the position of the MT seam can be pre-determined by the orientation of the γ -TuRC template, even before MT nucleation starts" as they don't directly show which protofilaments the seam forms between. They just show that it would take less movement to form between protofilaments 2 and 14.

Based on our reconstruction of the γ -TuRC-capped microtubule minus end, we can narrow down seam formation to either side of the protofilament bound to γ -TuRC spoke 2. Thus, the orientation of the γ -TuRC can pre-determine the position of the seam in relation to the cellular surroundings accurately.

We will expand the respective section in the manuscript to clarify this.

Minor comments

1) In the introduction, the authors say: "In vivo, however, MT nucleation is templated by γ -tubulin complexes (γ -TuCs), yielding MTs with a defined number of 13 protofilaments (Tilney et al, 1973)." This is an overstatement. Firstly, many cells have been shown to have microtubules with different protofilament numbers (see Review by Gary Brouhard). Secondly, the way the statement is made suggests that we already know that γ -TuRC-mediated nucleation produces 13-protofilament microtubules. But at this stage in the paper, we don't know whether the γ -TuRC will change conformation during nucleation and so could in theory produce a 14-protofilament microtubule.

We agree with the reviewer and we will tone down this statement in the introduction section of our manuscript.

2) In the introduction, the authors mention GCP2/Mzt2 modules and say that they have only been observed when bound to the CM1 domain. Firstly, is it known that this module binds the CM1 domain, or are they just close in space? Secondly, the authors should mention the work from the Luders lab that provides good evidence for Mzt2-GCP2-NTE modules decorating the outside of the γ -TuRC at GCP2/3 interfaces (Zimmerman et al).

Direct interaction between the GCP2/MZT2 module and the CM1 motif of CDK5RAP2 has been structurally resolved in the high-resolution reconstruction of the human γ -TuRC isolated via recombinant CM1-motif¹.

We will add this to the introduction section of our manuscript and also include a comment on the evidence for GCP2/MZT2 module decoration on the outside of the γ -TuRC from the Luders lab.

[1] Wieczorek M, Huang TL, Urnavicius L, Hsia KC, Kapoor TM (2020) MZT Proteins Form Multi-Faceted Structural Modules in the γ -Tubulin Ring Complex. *Cell Rep* 31: 107791

Referee #2 (Report for Author)

Microtubule (MT) dynamics are centrally involved in numerous cell activities, and the regulated addition and loss of tubulin subunits at both MT plus and minus ends is thus a critical aspect of cell physiology. A molecular level understanding of the architecture of MT ends, and of regulators of end dynamics, therefore provides important mechanistic insight into cellular processes. gamma-Tubulin Ring Complexes (g-TuRCs) both nucleate MTs, cap and stabilize MT minus ends, and their distribution within cells defines the organization of the MT array.

New information has recently been revealed about the structures of inactive g-TuRCs from a number of different eukaryotes using cryo-EM. As well as enabling direct and detailed visualization of these multi-protein complexes, this important set of studies provided suggestive insights as to how MT nucleation might occur on activation. However, precise experimental data about the structure of MT minus ends capped by g-TuRCs has been missing.

The study by Vermeulen et al seeks to address this knowledge gap. The authors mainly use tubulin-rich *Xenopus* egg extract to purify g-TuRC-nucleated MTs, which are drug-stabilized by either taxol or docetaxel. The purification protocol involves both "mild mechanical shearing" of longer MTs in the sample before affinity purification of the g-TuRCs (and attached shorter MTs) and subsequent cryo-EM grid preparation. It is therefore likely that these stabilizing drugs were important during sample preparation to maintain the integrity of the otherwise dynamic MTs. Because of challenges with the sample, the authors present an analysis of modestly-sized datasets of taxol- or docetaxel-stabilized MTs, and ultimately (and apparently legitimately) merge these datasets to achieve their final reconstruction of the g-TuRC with attached MT.

Albeit visualized at modest resolution (~20 Å), the authors' structure very surprisingly, does not display an exact match between the g-TuRC and the pseudo-helical structure of the MT that emerges from it. Rather, a patch of structural discontinuities is observed centred around g-tubulins 1 and 14, which do not perfectly align, and which renders the MT minus end slightly non-cylindrical, or as the authors describe it "profoundly asymmetric".

We will tone down this statement in the revised manuscript.

2 key questions arise from this finding which the authors need to address:

- 1) Does the reported asymmetric structure reflect the active *Xenopus* g-TuRCs in their samples?
- 2) What does the reported structure tell us about MT nucleation and capping mechanisms?

Question 1

The image processing scheme used by the authors involves inclusion of images of inactive g-TuRCs to drive 3D alignment. This is a counter-intuitive strategy given the aim of determining the structure of active g-TuRC. According to the authors' account, it seems that without the probably more overtly asymmetric inactive g-TuRC data, the small dataset of more symmetric active g-TuRCs would not align convergently. The inactive g-TuRC data is not included in the final reconstruction, and some evidence is provided (EV Fig 4C) that at least some crude

features of the inactive g-TuRC data are not carried through into the final active g-TuRC + MT reconstruction. However, concerns remain that the asymmetry of the inactive g-TuRC data has been imprinted on the final reconstruction such that reported asymmetry of the MT end arises from it.

We disagree with the reviewer on this point. We have thoroughly validated in a series of *in silico* experiments that the structure and conformation of supplemented particles did not influence the density observed for MT-capping γ -TuRCs.

In the first approach, density for the GRIP2 domains and γ -tubulin molecules in spokes 5 and 6 was subtracted from the supplemented particles and the initial reference before particle refinement (EV Fig. 4C). While these segments of spokes 5 and 6 are missing in the reconstruction of supplemented particles after joint refinement, as expected, they are clearly resolved and well covered in the reconstruction of the MT-capping γ -TuRCs, demonstrating that particles were correctly aligned without structural bias towards the supplemented particles or reference.

In the second approach, we performed a series of cross-correlation experiments, systematically comparing cryo-EM reconstructions of the MT-capping γ -TuRC obtained under varying particle refinement conditions with models of the 'open', the hypothetical 'closed' and the 'partially closed' γ -TuRC (EV Tables 2-4). These experiments established that the overall conformation in the reconstruction of the MT-capping γ -TuRC is not influenced by the supplemented particles.

Finally, even refinement runs in the absence of any supplemented particles result in reconstructions of a partially closed γ -TuRC (EV Fig. 5), although at lower resolution. This clearly demonstrates that the partially closed γ -TuRC conformation does not result from a bias introduced by the supplemented particles in an open conformation.

There is a hint this this might be the case from the data EV Fig 8B, in which more symmetric regions of the capped minus end are said to be more plastic, but this could equally be because they are less well aligned.

The data presented in EV Fig. 8 by no means provide a hint of less-well aligned γ -TuRC particles. On the contrary, the γ -TuRC itself is well resolved throughout, which is indicative of well aligned γ -TuRC-particles. Instead, the data rather illustrate that the microtubule segment associated with the symmetric half of the γ -TuRC (not the γ -TuRC itself) is less well resolved, which indicates conformational plasticity of the γ -TuRC-microtubule interaction.

We will clarify this in the revised version of the manuscript.

EV Fig 5 is also presented as evidence of a lack of bias in the 3D reconstruction, but if I have understood the data, this figure simply confirms that the presented reconstruction does not match either models of a hypothetical symmetric activated g-TuRC or of inactive g-TuRC.

In EV Fig. 5, we show a reconstruction that is different from the reconstruction analyzed in detail in the main figures (Figs. 2,3). The reconstruction in EV Fig. 5 was obtained in a refinement run at global angular sampling excluding any supplemented particles. Thus

importantly, although at lower resolution, this reconstruction demonstrates that a partially closed conformation of γ -TuRC is obtained even in the absence of supplemented γ -TuRC particles in the open conformation.

In an effort to entirely exclude supplemented γ -TuRC particles in the open conformation from the cryo-EM image processing scheme, we are in the process of extending the number of γ -TuRC-capped minus ends from 20,000 additional micrographs that have already been acquired. We expect the analysis to be concluded by end of December.

Question 2

Given the unexpected mismatch between the g-TuRC template and the nucleated MT, the observed structure does not lead straightforwardly to mechanistic clarity concerning precise nucleation of 13 protofilament MT and capping activity. The authors propose that additional regulatory steps may exist, and don't exclude that these could be additional steps that induce full symmetrization of the g-TuRC. However, it also remains a formal possibility that the mechanical shearing step in the purification protocol could also contribute to the slightly distorted MT minus end.

If shearing forces during microtubule shortening were to impact on the association between γ -TuRC and microtubule minus ends, the expected outcome would not be a homogenous structure with defined asymmetric protofilament organization, but rather a very heterogeneous mix of different minus end structures at different degrees of γ -TuRC detachment.

Moreover, shearing forces are expected to be maximal in the middle of microtubule segments, where the microtubule lattice is very well defined and ordered (EV Fig. 3).

In an effort to determine the structure of γ -TuRC-capped minus ends purified without any shortening procedure, thus circumventing shearing forces, we are in the process of analyzing γ -TuRC-capped minus ends from 12,000 additional micrographs that have already been acquired. The increased length of the γ -TuRC-associated microtubules poses challenges in terms of particle distribution and density on the cryo-EM grids, but we expect the analysis to be concluded by end of December.

Furthermore:

i) Both active yeast (<https://doi.org/10.21203/rs.3.rs-3481382/v1>) and human (<https://doi.org/10.1101/2023.11.20.567916>) g-TuRCs have been recently described at relatively higher resolution as forming symmetric structures at the minus ends of MTs. The major discrepancy between these studies and that of Vermeulen et al needs to be accounted for.

The yeast microtubule nucleation system analyzed in the first preprint manuscript structurally vastly differs from the vertebrate system. *S. cerevisiae* encodes only for a highly reduced set of relevant proteins (GCP2, GCP3, γ -tubulin) that form 2-spoked γ -Tubulin Small Complex (γ -TuSC) units. Oligomerization of seven identical γ -TuSCs forms the yeast γ -TuRC. By virtue of this uniform composition, the *S. cerevisiae* γ -TuRC is symmetrical already without

a microtubule minus end associated with it². Thus, very much different from vertebrate systems, a fully closed structure of the minus end-capping γ -TuRC was expected.

In the second preprint manuscript mentioned by the reviewer, a minimal *in vitro* microtubule nucleation system was used. The system included recombinant γ -TuRC of which the post-translational modification state is unclear, mutant β -tubulin with attenuated GTP hydrolysis kinetics and only one specific β -tubulin isotype (TUBB3) with non-canonical oligomerization properties [3]. Moreover, the microtubule nucleation system did not include a full physiological set of regulatory proteins of γ -TuRC-based microtubule nucleation. Thus, the composition of the *in vitro* microtubule nucleation system may converge into a microtubule nucleation pathway different from the one we studied and result in a fully closed structure of the minus end-capping γ -TuRC.

By contrast, we use the well-established *X. laevis* egg extract system with a physiological mixture of unmutated tubulin isoforms, native post-translational modifications as well as all the endogenous regulatory proteins. We induce microtubule nucleation by the addition of RanGTP, a physiological trigger of branching microtubule nucleation during spindle formation.

As outlined in our response to comment 1 of Reviewer #1, in the revised manuscript we will thoroughly discuss how our study relates to alternative microtubule nucleation pathways and we will include references to these two preprint manuscripts.

[2] Brilot AF, Lyon AS, Zelter A, Viswanath S, Maxwell A, MacCoss MJ, Muller EG, Sali A, Davis TN, Agard DA (2021) CM1-driven assembly and activation of yeast γ -tubulin small complex underlies microtubule nucleation. *eLife* 10: e65168

[3] Ti SC, Alushin GM, Kapoor TM (2019) Human β -tubulin isoforms can regulate microtubule protofilament number and stability. *Dev Cell* 47: 175-190

ii) The experiments relating to CAMSAP binding relate in part to recombinant human γ -TuRC about which no 3D structural information is provided. The authors can only infer that preferential CAMSAP binding is mediated via the asymmetric feature in their reconstruction. In addition, this model of CAMSAP recognition is incompletely reconciled with more detailed existing models of CAMSAP minus end binding e.g. PMID 28991265

The asymmetry and deformed protofilament organization in γ -TuRC-capped microtubule minus ends most likely reflect structural properties of uncapped microtubule minus ends that promote CAMSAP2 binding, as defined in the article highlighted by the reviewer (PMID 28991265).

However, we agree that our CAMSAP2 localization experiments do not directly demonstrate such specific recognition of asymmetric features by CAMSAP2 in the context of the γ -TuRC-capped microtubule minus end. Thus, in the revised version of the manuscript, we will emphasize that specific recognition of the γ -TuRC-capped minus end by CAMSAP2 via microtubule asymmetry and deformed protofilaments organization is a likely scenario, but

remains speculative. In line, we are considering to move Fig. 4, focusing on the CAMSAP2 binding experiment, to the supplements.

Independent of the exact mechanism of CAMSAP2 recruitment, the accumulation of CAMSAP2 at the γ -TuRC-capped microtubule minus end is an important finding that allows us to propose model for γ -TuRC turnover.

Please also see our response to Reviewer #3, comment 4.

Overall, results of this study are well presented and clearly described but there are number of critical aspects of study design and interpretation that urgently require clarification.

Referee #3 (Report for Author)

In recent years, several research groups, including those contributing to this manuscript, have elucidated the structure of the microtubule nucleator γ -tubulin ring complex (γ -TuRC) using cryo-electron microscopy (cryo-EM). The asymmetric nature of the γ -TuRC structure has prompted investigations into the regulation of this asymmetry in guiding the formation of symmetric microtubules. In this study, Vermeulen and colleagues address this question by reconstituting native γ -TuRC-capped microtubules from *Xenopus laevis* egg extract for cryo-EM analysis. The manuscript introduces a protocol for utilizing endogenous protein components of *Xenopus* egg extract to reconstitute γ -TuRC-templated microtubule nucleation through the RanGTP pathway.

Subsequent cryo-EM analysis of the γ -TuRC-capped microtubules at 17 Å reveals a partially closed ring geometry that misaligns with the emanating microtubule, deviating from the symmetry of a typical 13-protofilament microtubule. Furthermore, the γ -TuRC bound to the microtubule minus end adopts a destabilized luminal bridge, as indicated by the obscure density of the β -actin molecule. To elucidate the significance of this conformation, the authors demonstrate the interaction between CAMSAP2 and the minus end of the γ -TuRC-capped microtubule.

Increasing our comprehension of the structural compatibility of γ -TuRC as a template for microtubule nucleation and its significance in microtubule minus-end modulation, this manuscript offers valuable insights. However, before publication, we suggest addressing the following points:

1. Purification of γ -TuRC-Capped Microtubule Minus Ends: The authors employed mechanical force (extensive pipetting as described in the method section) to fragment long γ -TuRC-capped microtubules during the purification process. It is crucial to consider the potential impact of this shearing force on the structure of γ -TuRC and its capped microtubules. While the authors have demonstrated the structural integrity of the microtubule lattice away from the minus end, the preservative effect may be attributed to microtubule-specific stabilizing reagents (Paclitaxel or docetaxel), leaving the γ -TuRC-capped minus end susceptible to mechanical force.

Please see our response to comment 2 of Reviewer #2.

2. RanGTP-induced microtubule formation is associated with microtubule branching, as indicated by the presence of NEDD1 and the augmin complex in the isolated γ -TuRC-capped microtubules (Expanded view table 1). This raises the possibility that the partially closed γ -TuRC conformation represents the structure unique for NEDD1/augmin-bound γ -TuRC and plays a role in microtubule branching. A more insightful discussion on these findings would enhance the manuscript.

Please see our response to comment 1 of Reviewer #1.

3. While the resolution of the final density map allows rigid body fitting of the *Xenopus* γ -

TuRC model, Figure 2E and Expanded view Figure 7F do not clearly illustrate the fitting of MZT1/GCP6 N terminus and MZT1/GCP3 N terminus models into the density map. This ambiguity challenges the validation of the conclusion that MZT1 was well-resolved in this study.

Fig. 2E and EV Fig. 7F show zooms on the luminal bridge area in our reconstruction of the microtubule minus end-associated γ -TuRC. In both panels, the MZT1/GCP6 and MZT1/GCP3 modules are well visible and encompassed by the cryo-EM density envelope, demonstrating that these areas are well resolved.

For clarity, we will specifically color and label both of the MZT1 modules in the figure panels. Moreover, for better visualization, we will add a rotated view on the luminal bridge area with the fitted atomic model to the figures.

4. When reconstituting human γ -TuRC-capped microtubules in vitro for the CAMSAP2 localization assay (Figure 4A), caution is advised in assuming that the recombinant human γ -TuRC adopts a similar partially closed geometry as observed in Xenopus γ -TuRC-capped microtubules.

We agree that the experimental setup used for microtubule nucleation may impact on the conformation of the microtubule minus end-associated γ -TuRC.

To address this uncertainty, we are currently setting up experimental conditions for CAMSAP2 localization experiments using γ -TuRC-capped microtubules that have been nucleated using the same protocol as used for cryo-EM analysis. This experiment will allow us to dissect CAMSAP2 distribution along microtubules with established minus end asymmetry. Since CAMSAP2 localization experiments and data analysis have been streamlined in our group, we expect the experiments to be concluded by end of December.

Additionally, considering CAMSAP2's known binding and stabilization of free microtubule minus-ends, it may be premature to propose that CAMSAP2 recognizes the asymmetric conformation of γ -TuRC-capped microtubule minus-ends.

Please see our response to comment 2, ii), of Reviewer #2.

Minors:

1. Figure 4D presents negative staining EM micrographs illustrating microtubule minus ends capped by γ -TuRC. Unfortunately, the observation of capped minus ends is hindered by protein aggregation, making it challenging for readers. The authors should consider replacing these micrographs with higher-quality images to enhance visual interpretation.

The protein aggregates are caused by the antibodies that bind to CAMSAP2-GFP. Thus, we cannot bypass this problem since it is an inherent aspect of the experimental setup.

2. In Expanded view Figure 10A, it would be helpful to include a micrograph of the entire negative staining EM grid alongside the 2D averaged classes. This will offer readers a broader perspective on the experimental setup and facilitate a more comprehensive interpretation of the results.

We will include an overview micrograph, illustrating particle distribution and overall appearance of the sample.

Prof. Elmar Schiebel
Universität Heidelberg
Zentrum für Molekulare Biologie
Im Neuenheimer Feld 282
Heidelberg 69120
Germany

14th Dec 2023

Re: EMBOJ-2023-116073

γ -tubulin ring complex asymmetry induces local protofilament mismatch at the microtubule minus end

Dear Elmar and Stefan,

Thank you again for sending your tentative point-by-point response, and talking me through particular points raised in the referee reports on your manuscript on native frog γ -tubulin ring complex structure. I was happy to hear that the key concerns linked to sample purification and structure determination/model building can be clarified in a straightforward manner, and would in this light invite you to prepare and resubmit a manuscript revised along the lines proposed in your draft response. Please do keep me updated in case of unexpected problems with the revision work, or in case you should need an extension of the default three-month revision period after all.

I should remind you that our policy to allow only a single round of (major) revision makes it important to carefully answer all points raised by the referees at this stage. As always, competing manuscript published during the course of your revision will have no negative impact on our final decision on your study. Finally, please note the additional information and more detailed guidelines on how to prepare a revision below (and in our online Guide to Authors) - closely adhering to them shall greatly facilitate the editorial processing at the time of resubmission.

Thank you again for the opportunity to consider this work, and I look forward to receiving your revision in due time.

With best regards,

Hartmut

4) Each main and each Expanded View (EV) figure should be uploaded as individual production-quality files (preferably in .eps, .tif, .jpg formats). For suggestions on figure preparation/layout, please refer to our Figure Preparation Guidelines:

9) Digital image enhancement is acceptable practice, as long as it accurately represents the original data and conforms to community standards. If a figure has been subjected to significant electronic manipulation, this must be clearly noted in the figure legend and/or the 'Materials and Methods' section. The editors reserve the right to request original versions of figures and the original images that were used to assemble the figure. Finally, we generally encourage uploading of numerical as well as gel/blot image source data; for details see: embopress.org/page/journal/14602075/authorguide#sourcedata

At EMBO Press, we ask authors to provide source data for the main manuscript figures. Our source data coordinator will contact you to discuss which figure panels we would need source data for and will also provide you with helpful tips on how to upload and organize the files.

In the interest of ensuring the conceptual advance provided by the work, we recommend submitting a revision within 3 months (13th Mar 2024). Please discuss the revision progress ahead of this time with the editor if you require more time to complete the revisions. Use the link below to submit your revision:

Link Not Available

Point-by-Point Reply to Reviewers

We thank the reviewers for their thoughtful and supportive comments, considering this “an excellent paper” that offers “valuable insights” and “makes a conceptual advance to the field”. Below we address the specific points raised by the reviewers and elaborate on the corresponding changes in the manuscript.

Referee #1 (Report for Author)

In this paper, Vermeulen et al. present a 17 Angstrom resolution structure of the *Xenopus* γ -TuRC associated with the minus end of a nucleated microtubule. This goes beyond what has been achieved previously in higher Eukaryotes, where structures of unattached γ -TuRCs have been elucidated. These unattached γ -TuRCs were shown to have a semi-open conformation where γ -tubulin 1 and 14 did not overlap and where the helical pitch did not match that of a microtubule. Thus, an important question has remained as to whether γ -TuRCs change conformation during or before nucleation. To address this, the authors stimulated the nucleation process within *Xenopus* egg extracts by adding constitutively active RanGTP and then affinity-purified γ -TuRCs bound to microtubules. The authors surprisingly find that the microtubule-attached γ -TuRCs do not fully "close" resulting in mismatches between γ -tubulin and microtubule protofilaments. The largest mismatches occur at positions 2-7. Moreover, the protofilament from spoke 2 has clear gaps between the protofilaments coming from spokes 3 and 14, except for a small connection to protofilament 14 that may be via the GCP3/Mzt1 module previously observed in a similar position. The authors suggest that the asymmetry in protofilaments may provide selective binding for CAMSAP2, potentially explaining how it may recognise minus ends capped by γ -TuRCs, although this was not formally tested. The authors also suggest that the imperfect match allows tubulin concentration to have a greater impact on the nucleation process. The authors explained that there were a limited number of particles to analyse due to the native source of the γ -TuRCs, and so they used a "tailored multistep approach for data processing". This reviewer is not an expert in cryo-EM single particle analysis and so cannot properly comment on the validity of this approach, although from a non-expert view it seems reasonable and necessary. Overall, this is an excellent paper and makes a conceptual advance to the field.

We thank the reviewer for this very positive evaluation of our manuscript.

Major comments

1) The mass spec data shows that Augmin/HAUS is associated with the purified γ -TuRC-microtubule complexes. It was recently shown that Ran-GTP stimulates Augmin/HAUS-mediated microtubule nucleation (Kraus et al., 2023, JBC). Thus, it seems that the pathway of nucleation the authors are studying is Augmin-mediated. This should be explicitly stated in the results and perhaps discussed in terms of how this influences γ -TuRC conformation and activity, noting that a couple of papers have already concluded that Augmin binding increases γ -TuRC activity.

We agree with the reviewer that we may have captured the structure of a γ -TuRC-capped microtubule minus end specific for a Ran-GTP/Augmin-dependent microtubule branching pathway during spindle formation. In this pathway, thousands of microtubules are formed

within a short period of time, so a mismatch between γ -TuRC and nucleated microtubules may generate a defined breaking point that promotes fast turnover of γ -TuRCs.

Importantly, we cannot exclude that γ -TuRC-capped microtubule minus ends nucleated in alternative pathways (e.g. from centrosomes) may structurally differ to accommodate differing requirements in γ -TuRC turnover and minus end stabilization. For instance, the γ -TuRC may fully close in scenarios where stable and permanent binding to the newly nucleated microtubule minus end is required.

We have made the following changes in the manuscript to address these aspects:

- 1) We have modified the manuscript title: " γ -TuRC asymmetry induces local protofilament mismatch at the RanGTP-stimulated microtubule minus end"
- 2) We have specifically introduced the RanGTP/Augmin-dependent nucleation pathway in the introduction section and emphasized differences towards centrosomal microtubule nucleation.
- 3) We have included this aspect in an extended discussion section.
- 4) We have added additional emphasis to the origin of the analyzed γ -TuRC-capped microtubules from the RanGTP/Augmin-dependent nucleation pathway in *X. laevis* throughout the manuscript.

2) Similar to point 1, the mass spec data also shows the absence of CDK5RAP2 (CM1) domain. This is an important point, and, while touched upon in the discussion, I think the authors need to make this clear to the reader in the results section.

The absence of CDK5RAP2 in the mass spec data goes hand in hand with the possibility of having captured the structure of a γ -TuRC-capped microtubule minus end nucleated in a native microtubule branching pathway during spindle formation, in which CDK5RAP2 is not involved. We have emphasized this aspect in the results section of the revised manuscript:

"Consistently, mass spectrometry analysis identified α/β -tubulin as well as γ -TuRC components and interacting proteins in the sample (Appendix Table S1, full list available as source data), including components involved in RanGTP/Augmin-dependent MT nucleation, such as TPX2, XMAP215 and the Augmin subunits HAUS1-8. Consistent with the absence of centrosomes in *X. laevis* egg extract (Gruss, 2018), we did not find the CM1-containing protein CDK5RAP2, which activates centrosomal MT nucleation. This confirms that purified MT minus ends were nucleated in a native RanGTP-dependent spindle assembly pathway."

We have furthermore extended the discussion to reflect this aspect.

3) When the authors analyse the seam of the microtubule, they show the microtubule lattice >100nm from the γ -TuRC, stating that the microtubule lattice is regular. But do they know that the seam has formed correctly? They don't appear to look at the seam directly.

A properly formed seam is a geometrical prerequisite for the formation of a regular 13 protofilament microtubule lattice. Although we cannot directly identify the seam at the resolution achieved in our reconstruction of the microtubule lattice > 100 nm away from the minus end, we know that it must have formed.

Moreover, I don't see how this shows that their "results suggest that the position of the MT seam can be pre-determined by the orientation of the γ -TuRC template, even before MT nucleation starts" as they don't directly show which protofilaments the seam forms between. They just show that it would take less movement to form between protofilaments 2 and 14.

Based on our reconstruction of the γ -TuRC-capped microtubule minus end, we can narrow down seam formation to either side of the protofilament bound to γ -TuRC spoke 2. Thus, the orientation of the γ -TuRC can pre-determine the position of the seam in relation to the cellular surroundings accurately.

We have expanded the respective section in the manuscript to clarify this:

"Altogether, our results suggest that the MT seam forms at either side of the protofilament at spoke 2 and thereby, the orientation of the γ -TuRC template can pre-determine the position of the MT seam, even before MT nucleation starts. The asymmetric order of GCP subunits within the γ -TuRC is ideally suited for positioning the γ -TuRC in a defined spatial orientation with respect to its cellular surroundings to predefine the seam."

Minor comments

1) In the introduction, the authors say: "In vivo, however, MT nucleation is templated by γ -tubulin complexes (γ -TuCs), yielding MTs with a defined number of 13 protofilaments (Tilney et al, 1973)." This is an overstatement. Firstly, many cells have been shown to have microtubules with different protofilament numbers (see Review by Gary Brouhard). Secondly, the way the statement is made suggests that we already know that γ -TuRC-mediated nucleation produces 13-protofilament microtubules. But at this stage in the paper, we don't know whether the γ -TuRC will change conformation during nucleation and so could in theory produce a 14-protofilament microtubule.

We agree with the reviewer and we have toned down this statement in the introduction section of our manuscript.

"*In vivo*, however, MT nucleation is templated by γ -tubulin complexes (γ -TuCs), allowing for spatiotemporal regulation of MT nucleation and subsequent MT minus-end anchoring (Wiese & Zheng, 2000). Additionally, this potentially enables γ -TuCs to determine the protofilament number of the nucleated MT, which is 13 in the vast majority of situations (Chaaban & Brouhard, 2017; Tilney et al, 1973)."

2) In the introduction, the authors mention GCP2/Mzt2 modules and say that they have only been observed when bound to the CM1 domain. Firstly, is it known that this module binds the CM1 domain, or are they just close in space? Secondly, the authors should mention the work from the Luders lab that provides good evidence for Mzt2-GCP2-NTE modules decorating the outside of the γ -TuRC at GCP2/3 interfaces (Zimmerman et al).

Direct interaction between the GCP2/MZT2 module and the CM1 motif of CDK5RAP2 has been structurally resolved in high-resolution reconstructions of vertebrate γ -TuRCs isolated via the recombinant CM1-motif of CDK5RAP2^{1,2}.

We have added this to the introduction section of our manuscript and also included a comment on the evidence for GCP2/MZT2 module decoration on the outside of the γ -TuRC from the Lüders lab:

“MZT2, on the other hand, forms a structural module with the N-terminus of GCP2, which has thus far only been observed in cryo-EM reconstructions when stably docked to the γ -TuRC in complex with the CM1 motif of CDK5RAP2 (Wieczorek *et al.*, 2020a; Wieczorek *et al.*, 2020b; Xu *et al.*, 2023). Crosslinks from the MZT2/GCP2^N module map to the outside surface of GCP2/3 (Zimmermann *et al.*, 2020), suggesting that this module primarily samples positions on the outside of the γ -TuRC even in the absence of CDK5RAP2.”

[1] Wieczorek M, Huang TL, Urnavicius L, Hsia KC, Kapoor TM (2020) MZT Proteins Form Multi-Faceted Structural Modules in the γ -Tubulin Ring Complex. *Cell Rep* 31: 107791

[2] Xu Y, Muñoz-Hernández H, Krutyhołowa R, Marxer F, Cetin F, Wieczorek M (2023) Closure of the γ -tubulin ring complex by CDK5RAP2 activates microtubule nucleation. *BioRxiv* 2023.12.14.571518; <https://doi.org/10.1101/2023.12.14.571518>

Referee #2 (Report for Author)

Microtubule (MT) dynamics are centrally involved in numerous cell activities, and the regulated addition and loss of tubulin subunits at both MT plus and minus ends is thus a critical aspect of cell physiology. A molecular level understanding of the architecture of MT ends, and of regulators of end dynamics, therefore provides important mechanistic insight into cellular processes. gamma-Tubulin Ring Complexes (g-TuRCs) both nucleate MTs, cap and stabilize MT minus ends, and their distribution within cells defines the organization of the MT array.

New information has recently been revealed about the structures of inactive g-TuRCs from a number of different eukaryotes using cryo-EM. As well as enabling direct and detailed visualization of these multi-protein complexes, this important set of studies provided suggestive insights as to how MT nucleation might occur on activation. However, precise experimental data about the structure of MT minus ends capped by g-TuRCs has been missing.

The study by Vermeulen et al seeks to address this knowledge gap. The authors mainly use tubulin-rich *Xenopus* egg extract to purify g-TuRC-nucleated MTs, which are drug-stabilized by either taxol or docetaxel. The purification protocol involves both "mild mechanical shearing" of longer MTs in the sample before affinity purification of the g-TuRCs (and attached shorter MTs) and subsequent cryo-EM grid preparation. It is therefore likely that these stabilizing drugs were important during sample preparation to maintain the integrity of the otherwise dynamic MTs. Because of challenges with the sample, the authors present an analysis of modestly-sized datasets of taxol- or docetaxel-stabilized MTs, and ultimately (and apparently legitimately) merge these datasets to achieve their final reconstruction of the g-TuRC with attached MT.

Albeit visualized at modest resolution (~20 Å), the authors' structure very surprisingly, does not display an exact match between the g-TuRC and the pseudo-helical structure of the MT that emerges from it. Rather, a patch of structural discontinuities is observed centred around g-tubulins 1 and 14, which do not perfectly align, and which renders the MT minus end slightly non-cylindrical, or as the authors describe it "profoundly asymmetric".

We have toned down this statement in the revised manuscript.

2 key questions arise from this finding which the authors need to address:

- 1) Does the reported asymmetric structure reflect the active *Xenopus* g-TuRCs in their samples?
- 2) What does the reported structure tell us about MT nucleation and capping mechanisms?

Question 1

The image processing scheme used by the authors involves inclusion of images of inactive g-TuRCs to drive 3D alignment. This is a counter-intuitive strategy given the aim of determining the structure of active g-TuRC. According to the authors' account, it seems that without the probably more overtly asymmetric inactive g-TuRC data, the small dataset of more symmetric active g-TuRCs would not align convergently. The inactive g-TuRC data is not included in the final reconstruction, and some evidence is provided (EV Fig 4C) that at least some crude

features of the inactive g-TuRC data are not carried through into the final active g-TuRC + MT reconstruction. However, concerns remain that the asymmetry of the inactive g-TuRC data has been imprinted on the final reconstruction such that reported asymmetry of the MT end arises from it.

We strongly disagree with the reviewer on this point. We have thoroughly validated in a series of *in silico* experiments that the structure and conformation of supplemented particles did not influence the density observed for MT-capping γ -TuRCs.

In the first approach, density for the GRIP2 domains and γ -tubulin molecules in spokes 5 and 6 was subtracted from the supplemented particles and the initial reference before particle refinement (EV Fig. 4C; EV Fig. 1C in the revised manuscript). While these segments of spokes 5 and 6 are missing in the reconstruction of supplemented particles after joint refinement, as expected, they are clearly resolved and well covered in the reconstruction of the MT-capping γ -TuRCs, demonstrating that particles were correctly aligned without structural bias towards the supplemented particles or reference.

In the second approach, we performed a series of cross-correlation experiments, systematically comparing cryo-EM reconstructions of the MT-capping γ -TuRC obtained under varying particle refinement conditions with models of the 'open', the hypothetical 'closed' and the 'partially closed' γ -TuRC (EV Table 1 and Appendix Tables S2-4 in the revised manuscript). These experiments established that the overall conformation in the reconstruction of the MT-capping γ -TuRC is not influenced by the supplemented particles.

Finally, even refinement runs in the absence of any supplemented particles result in reconstructions of a partially closed γ -TuRC (EV Fig. 2 in the revised manuscript), although at lower resolution. This analysis clearly demonstrates that the partially closed γ -TuRC conformation does not result from a bias introduced by the supplemented particles in an open conformation.

We have included a paragraph explicitly summarizing these validation experiments in the discussion section.

There is a hint this this might be the case from the data EV Fig 8B, in which more symmetric regions of the capped minus end are said to be more plastic, but this could equally be because they are less well aligned.

The data presented in EV Fig. 8 (EV Fig. 4 in the revised manuscript) by no means provide a hint of less-well aligned γ -TuRC particles. On the contrary, the γ -TuRC itself is well resolved throughout, which is indicative of well aligned particles. Instead, the data rather illustrate that the microtubule segment associated with the symmetric half of the γ -TuRC (not the γ -TuRC itself) is less well resolved, which indicates conformational plasticity of the γ -TuRC-microtubule interaction.

We have clarified this in the revised version of the manuscript:

"While γ -TuRC-derived density segments are well resolved in the reconstruction obtained after γ -TuRC-focused 3D refinement (Fig. 2A), the density representing the emanating MT is locally less well-defined (Fig. EV4), indicative of structural plasticity at the interface between γ -TuRC and MT."

EV Fig 5 is also presented as evidence of a lack of bias in the 3D reconstruction, but if I have understood the data, this figure simply confirms that the presented reconstruction does not match either models of a hypothetical symmetric activated g-TuRC or of inactive g-TuRC.

In EV Fig. 5 (EV Fig. 2 in the revised manuscript), we show a reconstruction that is different from the reconstruction analyzed in detail in the main figures (Figs. 2,3). The reconstruction in EV Fig. 5 (EV Fig. 2 in the revised manuscript) was obtained in a refinement run at global angular sampling excluding any supplemented particles. Thus importantly, although at lower resolution, this reconstruction demonstrates that a partially closed conformation of γ -TuRC is obtained even in the absence of supplemented γ -TuRC particles.

We have emphasized this in the results and discussion sections of the manuscript, as well as the figure legend of EV Fig. 5 (EV Fig. 2 in the revised manuscript).

Question 2

Given the unexpected mismatch between the g-TuRC template and the nucleated MT, the observed structure does not lead straightforwardly to mechanistic clarity concerning precise nucleation of 13 protofilament MT and capping activity. The authors propose that additional regulatory steps may exist, and don't exclude that these could be additional steps that induce full symmetrization of the g-TuRC. However, it also remains a formal possibility that the mechanical shearing step in the purification protocol could also contribute to the slightly distorted MT minus end.

Shearing forces are expected to be maximal in the middle of microtubule segments, where the microtubule lattice is well defined and ordered (EV Fig. 3; Appendix Fig. S3 in the revised manuscript).

However, in response to this comment and to entirely rule out that shearing forces originating from MT shortening during sample purification contributed to the misalignment and asymmetry of the γ -TuRC and its associated MT, we have analysed the structure of γ -TuRC-capped minus ends purified without any shortening procedure, thus circumventing shearing forces. The increased length of γ -TuRC-capped MTs posed challenges in terms of sample purification, as well as particle distribution and density on the cryo-EM grids. However, following the same approaches to cryo-EM data processing and structural analysis as before, we observed γ -TuRC and MT asymmetry indistinguishable from γ -TuRC-capped MTs to which the shortening procedure was applied (EV Fig. 5, Appendix Table S4, Appendix Fig. S6). This confirms that MT shortening had no effect on the MT minus end structure. We have included these data in the revised version of the manuscript.

Furthermore:

i) Both active yeast (<https://doi.org/10.21203/rs.3.rs-3481382/v1>) and human (<https://doi.org/10.1101/2023.11.20.567916>) g-TuRCs have been recently described at relatively higher resolution as forming symmetric structures at the minus ends of MTs. The major discrepancy between these studies and that of Vermeulen et al needs to be accounted for.

The yeast microtubule nucleation system analyzed in the first preprint manuscript structurally vastly differs from the vertebrate system. *S. cerevisiae* encodes only for a highly reduced set of relevant proteins (GCP2, GCP3, γ -tubulin) that form 2-spoked γ -Tubulin Small Complex (γ -TuSC) units. Oligomerization of seven identical γ -TuSCs forms the yeast γ -TuRC. By virtue of this uniform composition, the *S. cerevisiae* γ -TuRC is symmetrical already without a microtubule minus end associated with it³. Thus, very much different from vertebrate systems, a fully closed structure of the minus end-capping γ -TuRC was expected.

In the second preprint manuscript mentioned by the reviewer, a minimal *in vitro* microtubule nucleation system was used. The system included recombinant γ -TuRC of which the post-translational modification state is unclear, mutant α -tubulin which attenuates GTP hydrolysis kinetics and only one specific β -tubulin isotype (TUBB3) with non-canonical oligomerization properties⁴. Moreover, the microtubule nucleation system did not include a full physiological set of regulatory proteins of γ -TuRC-based microtubule nucleation. Thus, the composition of the *in vitro* microtubule nucleation system may converge into a microtubule nucleation pathway different from the one we studied and result in a fully closed structure of the minus end-capping γ -TuRC.

By contrast, we use the well-established *X. laevis* egg extract system with a physiological mixture of unmutated tubulin isoforms, native post-translational modifications as well as all the endogenous regulatory proteins. We induce microtubule nucleation by the addition of RanGTP, a physiological trigger of branching microtubule nucleation during spindle formation.

As outlined in our response to comment 1 of Reviewer #1, in the revised manuscript we have thoroughly discussed how our study relates to alternative microtubule nucleation pathways and we have included references to the two preprint manuscripts mentioned above, as well as a recently published article⁵.

[3] Brilot AF, Lyon AS, Zelter A, Viswanath S, Maxwell A, MacCoss MJ, Muller EG, Sali A, Davis TN, Agard DA (2021) CM1-driven assembly and activation of yeast γ -tubulin small complex underlies microtubule nucleation. *eLife* 10: e65168

[4] Ti SC, Alushin GM, Kapoor TM (2019) Human β -tubulin isoforms can regulate microtubule protofilament number and stability. *Dev Cell* 47: 175-190

[5] Brito C, Serna M, Guerra P, Llorca O, Surrey T (2024) Transition of human γ -tubulin ring complex into a closed conformation during microtubule nucleation. *Science*
<https://doi.org/10.1126/science.adk6160>

ii) The experiments relating to CAMSAP binding relate in part to recombinant human γ -TuRC about which no 3D structural information is provided. The authors can only infer that preferential CAMSAP binding is mediated via the asymmetric feature in their reconstruction. In addition, this model of CAMSAP recognition is incompletely reconciled with more detailed existing models of CAMSAP minus end binding e.g. PMID 28991265

The asymmetry and deformed protofilament organization in γ -TuRC-capped microtubule minus ends most likely reflect structural properties that promote CAMSAP2 binding, as

defined for uncapped microtubule minus ends in the article highlighted by the reviewer (PMID 28991265).

However, we agree that our CAMSAP2 localization experiments do not directly demonstrate such specific recognition of asymmetric features by CAMSAP2 in the context of the γ -TuRC-capped microtubule minus end. Thus, in the revised version of the manuscript, we have emphasized that specific recognition of the γ -TuRC-capped minus end by CAMSAP2 via microtubule asymmetry and deformed protofilaments organization is a likely scenario, but remains speculative.

Independent of the exact mechanism of CAMSAP2 recruitment, the accumulation of CAMSAP2 at the γ -TuRC-capped microtubule minus end is an important finding that allows us to propose a model for γ -TuRC turnover.

In addition, we have now repeated our CAMSAP2 localization experiments, but using γ -TuRC-capped microtubule minus ends purified from *X. laevis* egg extract in the same way as done for the cryo-EM analysis. In this experiment, we observed a similar preference for binding at the γ -TuRC-capped microtubule minus end.

Please also see our response to reviewer #3, comment 4.

Overall, results of this study are well presented and clearly described but there are number of critical aspects of study design and interpretation that urgently require clarification.

Referee #3 (Report for Author)

In recent years, several research groups, including those contributing to this manuscript, have elucidated the structure of the microtubule nucleator γ -tubulin ring complex (γ -TuRC) using cryo-electron microscopy (cryo-EM). The asymmetric nature of the γ -TuRC structure has prompted investigations into the regulation of this asymmetry in guiding the formation of symmetric microtubules. In this study, Vermeulen and colleagues address this question by reconstituting native γ -TuRC-capped microtubules from *Xenopus laevis* egg extract for cryo-EM analysis. The manuscript introduces a protocol for utilizing endogenous protein components of *Xenopus* egg extract to reconstitute γ -TuRC-templated microtubule nucleation through the RanGTP pathway.

Subsequent cryo-EM analysis of the γ -TuRC-capped microtubules at 17 Å reveals a partially closed ring geometry that misaligns with the emanating microtubule, deviating from the symmetry of a typical 13-protofilament microtubule. Furthermore, the γ -TuRC bound to the microtubule minus end adopts a destabilized luminal bridge, as indicated by the obscure density of the β -actin molecule. To elucidate the significance of this conformation, the authors demonstrate the interaction between CAMSAP2 and the minus end of the γ -TuRC-capped microtubule.

Increasing our comprehension of the structural compatibility of γ -TuRC as a template for microtubule nucleation and its significance in microtubule minus-end modulation, this manuscript offers valuable insights.

We thank the reviewer for this very positive evaluation of our manuscript.

However, before publication, we suggest addressing the following points:

1. Purification of γ -TuRC-Capped Microtubule Minus Ends: The authors employed mechanical force (extensive pipetting as described in the method section) to fragment long γ -TuRC-capped microtubules during the purification process. It is crucial to consider the potential impact of this shearing force on the structure of γ -TuRC and its capped microtubules. While the authors have demonstrated the structural integrity of the microtubule lattice away from the minus end, the preservative effect may be attributed to microtubule-specific stabilizing reagents (Paclitaxel or docetaxel), leaving the γ -TuRC-capped minus end susceptible to mechanical force.

Please see our response to comment 2 of Reviewer #2.

2. RanGTP-induced microtubule formation is associated with microtubule branching, as indicated by the presence of NEDD1 and the augmin complex in the isolated γ -TuRC-capped microtubules (Expanded view table 1). This raises the possibility that the partially closed γ -TuRC conformation represents the structure unique for NEDD1/augmin-bound γ -TuRC and plays a role in microtubule branching. A more insightful discussion on these findings would enhance the manuscript.

Please see our response to comment 1 of Reviewer #1.

3. While the resolution of the final density map allows rigid body fitting of the *Xenopus* γ -TuRC model, Figure 2E and Expanded view Figure 7F do not clearly illustrate the fitting of MZT1/GCP6 N terminus and MZT1/GCP3 N terminus models into the density map. This ambiguity challenges the validation of the conclusion that MZT1 was well-resolved in this study.

Fig. 2E and EV Fig. 7F (Appendix Fig. S4F in the revised manuscript) show zooms on the luminal bridge area in our reconstruction of the microtubule minus end-associated γ -TuRC. In both panels, the MZT1/GCP6 and MZT1/GCP3 modules are well visible and encompassed by the cryo-EM density envelope, demonstrating that these areas are well resolved.

For clarity, we have now specifically colored and labeled the atomic model for both of the MZT1 modules and actin in the figure panels, while the rest of the model was colored in grey.

4. When reconstituting human γ -TuRC-capped microtubules *in vitro* for the CAMSAP2 localization assay (Figure 4A), caution is advised in assuming that the recombinant human γ -TuRC adopts a similar partially closed geometry as observed in *Xenopus* γ -TuRC-capped microtubules.

We agree that the experimental setup used for microtubule nucleation may impact on the conformation of the microtubule minus end-associated γ -TuRC.

To address this, we have optimized experimental conditions for CAMSAP2 localization experiments using γ -TuRC-capped microtubules nucleated in *X. laevis* egg extract using the same protocol as used for cryo-EM analysis. Using this system, we observed a similar distribution of CAMSAP2 as seen before on γ -TuRC-capped microtubules nucleated *in vitro*. In particular, we observe strong enrichment of CAMSAP2 at the γ -TuRC-capped microtubule minus end as compared to the microtubule lattice or non-capped microtubule minus ends. This validates CAMSAP2 enrichment to γ -TuRC-capped microtubule minus ends with established minus end asymmetry. These data have been included in the manuscript in Fig. 4, whereas data previously included in Fig. 4 has been moved to Appendix Figs. S7 and S8 in the revised manuscript.

Additionally, considering CAMSAP2's known binding and stabilization of free microtubule minus-ends, it may be premature to propose that CAMSAP2 recognizes the asymmetric conformation of γ -TuRC-capped microtubule minus-ends.

Please see our response to comment 2, ii), of Reviewer #2.

Minors:

1. Figure 4D presents negative staining EM micrographs illustrating microtubule minus ends capped by γ -TuRC. Unfortunately, the observation of capped minus ends is hindered by

protein aggregation, making it challenging for readers. The authors should consider replacing these micrographs with higher-quality images to enhance visual interpretation.

The protein aggregates are caused by the antibodies that bind to CAMSAP2-GFP. Thus, we cannot bypass this problem since it is an inherent aspect of the experimental setup. Since, we now confirmed CAMSAP2-GFP localization at the capped microtubule minus ends purified from *X. laevis* egg extract by an optimized indirect immunofluorescence approach (Fig. 4), we moved the immunogold co-localization experiments to the appendix (Appendix Fig. S7).

2. In Expanded view Figure 10A, it would be helpful to include a micrograph of the entire negative staining EM grid alongside the 2D averaged classes. This will offer readers a broader perspective on the experimental setup and facilitate a more comprehensive interpretation of the results.

We have included an overview micrograph, illustrating particle distribution and overall appearance of the sample in Appendix Fig. S8B of the revised manuscript.

Prof. Elmar Schiebel
Universität Heidelberg
Zentrum für Molekulare Biologie
Im Neuenheimer Feld 282
Heidelberg 69120
Germany

11th Mar 2024

Re: EMBOJ-2023-116073R

γ -TuRC asymmetry induces local protofilament mismatch at the RanGTP-stimulated microtubule minus end

Dear Elmar and Stefan,

Thank you for submitting your revised manuscript. The original referees 2 and 3 have now assessed it once again, and provided the comments copied below. Since especially the image processing concerns appear to have been addressed to the referees' satisfaction, we shall be happy to proceed with acceptance and publication of the study, following incorporation of a few remaining editorial points:

- Please modify the abstract in line with referee 3's remaining comment, and please provide a final response letter (and, if needed, text changes) to referee 2's remaining reservations.
- Please check that all figure panels have been referenced at least once, e.g. callouts for Fig. 4A-C, 5B appear to be missing.
- Please upload Table EV1, together with its legend, as a separate DOCX file
- Please adjust the format for citation of preprints as specified in our author guidelines:
The citation in the text should be: " (preprint: NAME1 et al, YEAR) "
The citation in the reference list: " Author NAME1, Author NAME2, ... (YEAR) article title. bioRxiv/ResearchSquare doi: XXX"
- Please rename the Conflict of Interest section into "Disclosure and competing interests statement" as specified in our Guide to Authors - for details, see <https://www.embopress.org/competing-interests>
- As we are switching from a free-text author contribution statement towards a more formal statement based on Contributor Role Taxonomy (CRediT) terms, please remove the present Author Contribution section and instead specify each author's contribution(s) directly in the Author Information page of our submission system during upload of the final manuscript. See <https://casrai.org/credit/> for more information.
- Please make sure to fully complete the Author Checklist file (where for one or more points, a pulldown selection is still missing).
- Finally, during our routine pre-acceptance checks, our data editors have raised the following queries regarding figures, data, and legends:
*Although 'N' is provided, please describe the nature of entity for 'N' in the legend of figure 1d.
* Please note that the measure of center for the error bar needs to be defined in the legend of figure 1d.

I am therefore returning the manuscript to you for a final round of minor revision, to allow you to make these adjustments and upload all modified files. Once we will have received them, we should be ready to swiftly proceed with formal acceptance and production of the manuscript.

With kind regards,

Hartmut

*** PLEASE NOTE: All revised manuscript are subject to initial checks for completeness and adherence to our formatting

guidelines. Revisions may be returned to the authors and delayed in their editorial re-evaluation if they fail to comply to the following requirements (see also our Guide to Authors for further information):

9) Digital image enhancement is acceptable practice, as long as it accurately represents the original data and conforms to community standards. If a figure has been subjected to significant electronic manipulation, this must be clearly noted in the figure legend and/or the 'Materials and Methods' section. The editors reserve the right to request original versions of figures and the original images that were used to assemble the figure. Finally, we generally encourage uploading of numerical as well as gel/blot image source data; for details see: embopress.org/page/journal/14602075/authorguide#sourcedata

At EMBO Press, we ask authors to provide source data for the main manuscript figures. Our source data coordinator will contact you to discuss which figure panels we would need source data for and will also provide you with helpful tips on how to upload and organize the files.

In the interest of ensuring the conceptual advance provided by the work, we recommend submitting a revision within 3 months (9th Jun 2024). Please discuss the revision progress ahead of this time with the editor if you require more time to complete the revisions. Use the link below to submit your revision:

Link Not Available

Referee #2:

I appreciate that the authors have addressed a number of the earlier comments in their revised manuscript. While I remain overall sceptical about the apparent necessity of including images of inactive g-TuRCs to drive alignment of the capped microtubule minus ends, the authors have clearly presented their image processing strategy and readers will be able to draw their own conclusions.

The authors have also elaborated on discussion of their observed partially closed, asymmetric structure compared with other recent studies that report structures with a precise match between the nucleating g-TuRC and the nucleated 13-protofilament microtubule (Aher et al 2023, Barford et al 2023, Brito et al 2024). One argument put forward is that their partially closed structure is specific to the RanGTP pathway. However, since no additional components of the pathway are visualised that could provide the mechanistic basis for this hypothesis, no appropriate control structures in the absence of RanGTP pathway components are presented, and the authors note that Brito et al also captured a similar partially closed conformation, this argument is unconvincing. The authors also imply that since the structures of Aher et al 2023, Barford et al 2023, Brito et al 2024 are in vitro reconstituted, in contrast to the ex vivo *Xenopus* extract Vermeulen et al have used, the resulting structures may be artefactual because of missing components in the reconstitution experiments. They also suggest that the *Xenopus* g-TuRC could undergo further conformational changes to adopt the fully closed structure observed by others under other experimental conditions.

All of these ideas are in principle legitimate points for discussion, but the collective effect is of mechanistic vagueness that is unlikely to provide clarity for the field.

Referee #3:

In the revised manuscript, Vermeulen et al. have addressed the issues and concerns raised previously. However, a minor point still needs attention. Since the authors have revised the manuscript title to emphasize the RanGTP-stimulated microtubule minus end, they should also indicate the significance of the RanGTP-dependent microtubule nucleation pathway in the abstract.

Point-by-Point Reply to Reviewers

We thank the reviewers for their final comments on our revised manuscript. Below we address the specific points raised by the reviewers and elaborate on the corresponding changes in the manuscript.

Referee #2:

I appreciate that the authors have addressed a number of the earlier comments in their revised manuscript. While I remain overall sceptical about the apparent necessity of including images of inactive γ -TuRCs to drive alignment of the capped microtubule minus ends, the authors have clearly presented their image processing strategy and readers will be able to draw their own conclusions.

We appreciate the reviewer's comment. We have thoroughly validated our image processing workflow in an extensive series of *in silico* experiments that are clearly presented in the manuscript. We are therefore very confident that readers will be convinced by the strong data and will be able to draw an informed conclusion.

The authors have also elaborated on discussion of their observed partially closed, asymmetric structure compared with other recent studies that report structures with a precise match between the nucleating γ -TuRC and the nucleated 13-protofilament microtubule (Aher et al 2023, Barford et al 2023, Brito et al 2024). One argument put forward is that their partially closed structure is specific to the RanGTP pathway. However, since no additional components of the pathway are visualised that could provide the mechanistic basis for this hypothesis, no appropriate control structures in the absence of RanGTP pathway components are presented, and the authors note that Brito et al also captured a similar partially closed conformation, this argument is unconvincing.

We strongly disagree with the reviewer on this point. In *Xenopus laevis* egg extract, microtubule nucleation occurs predominantly via the RanGTP-induced spindle assembly pathway. Consistently, mass spectrometry analysis demonstrates that the γ -TuRC-capped microtubule minus ends used for cryo-EM analysis contain all components required for this pathway, while central components for centrosomal microtubule nucleation pathways are not present (Appendix Table 1). Thus, different from the other studies mentioned above, we clearly have visualized the microtubule minus ends natively nucleated by the RanGTP-induced spindle assembly pathway. Notably, the observation of a similar partially closed conformation by Brito et al merely underlines that the γ -TuRC can adopt such a conformation, but does not link it to any specific cellular pathway.

The authors also imply that since the structures of Aher et al 2023, Barford et al 2023, Brito et al 2024 are *in vitro* reconstituted, in contrast to the *ex vivo* *Xenopus* extract Vermeulen et al have used, the resulting structures may be artefactual because of missing components in the reconstitution experiments.

This statement is not correct. For none of the studies mentioned by the reviewer, we implied that it is artefactual.

- 1) Barford *et al*, 2023: We suggested that observing a fully symmetric γ -TuRC-capped microtubule minus end in *S. cerevisiae* was highly expected, because of the inherent symmetry of yeast γ -TuRC, consisting of seven identical copies of GCP2-GCP3 units. This situation vastly differs from the asymmetric conformation and molecular

architecture of the vertebrate γ -TuRC, containing different copy numbers of GCP4-6 in addition to GCP2/3, unevenly distributed within the complex.

- 2) Aher *et al*, 2023, Brito *et al*, 2024: We suggested that the different composition of the microtubule nucleation systems used in these two studies may have converged into a different microtubule nucleation pathway, as compared to our study.

They also suggest that the *Xenopus* γ -TuC could undergo further conformational changes to adopt the fully closed structure observed by others under other experimental conditions.

All of these ideas are in principle legitimate points for discussion, but the collective effect is of mechanistic vagueness that is unlikely to provide clarity for the field.

We disagree with this comment. Our study clearly links a defined microtubule minus end geometry to a defined microtubule nucleation pathway, i.e. a fully native, RanGTP-induced spindle assembly pathway.

Referee #3:

In the revised manuscript, Vermeulen *et al.* have addressed the issues and concerns raised previously. However, a minor point still needs attention. Since the authors have revised the manuscript title to emphasize the RanGTP-stimulated microtubule minus end, they should also indicate the significance of the RanGTP-dependent microtubule nucleation pathway in the abstract.

We have revised our abstract accordingly and extended two sentences to emphasize the significance of the RanGTP-dependent microtubule nucleation pathway with the text marked in bold:

Here, we isolate native γ -TuRC-capped microtubules from *Xenopus laevis* egg extract **nucleated through the RanGTP-induced pathway for spindle assembly** and determine their cryo-EM structure.

Collectively, we reveal a surprisingly asymmetric microtubule minus end protofilament organisation diverging from the regular microtubule structure, with direct implications for the kinetics and regulation of nucleation and subsequent modulation of microtubules **during spindle assembly**.

Prof. Elmar Schiebel
Universität Heidelberg
Zentrum für Molekulare Biologie
Im Neuenheimer Feld 282
Heidelberg 69120
Germany

18th Mar 2024

Re: EMBOJ-2023-116073R1

γ -TuRC asymmetry induces local protofilament mismatch at the RanGTP-stimulated microtubule minus end

Dear Elmar and Stefan,

Thank you for submitting your final revised manuscript for our consideration. I am pleased to inform you that we have now accepted it for publication in The EMBO Journal.

Yours sincerely,

Hartmut
